



# Coupled estimation of incoherent inertia gravity wave field and turbulent balanced motions via modal decomposition

Igor Maingonnat[1], Gilles Tissot[1,*], and Noé Lahaye*[1,*]

[1]INRIA Rennes Bretagne Atlantique, IRMAR – UMR CNRS 6625, av. General Leclerc, 35042 Rennes, France
[*]These authors contributed equally to this work.

**Correspondence:** Igor Maingonnat (igor.maingonnat@inria.fr)

**Abstract.** The non-linear interactions between an internal wave field propagating through a balanced turbulent flow make the incoherent propagation of ocean internal tides difficult to understand and to predict. In this paper, we propose a data-driven method that extracts the structures of a wave field correlated with the fluctuations of the background flow, exploiting the fact that the scattering of the wave results from a quadratic nonlinearity involving both types of motions. The method consists of applying the extended proper orthogonal decomposition method to the complex wave envelope, extracted by complex demodulation, in order to provide the spatial structures of an incoherent wave contribution correlated to the proper orthogonal decomposition modes of the balanced motion. Using the rotating shallow water as a canonical model for wave / balanced flow interactions, we describe the connection between the variability of the jet and the incoherence of the internal wave, which provides some theoretical foundations to the proposed method. This expected connection is then confirmed by numerical simulations of the rotating shallow water model, and the variability modes of the jet and wave field are analysed. Finally, an algorithm for estimating the jet component and the associated wave field from single snapshots of the sea level height is proposed and tested using outputs from numerical simulations. We show that such algorithm provides a valuable coupled estimate of the two dynamics, especially in configurations where the wave signal is small compared to the jet.

*Keywords:* Internal wave interactions, wave scattering, data-driven methods, estimation.

## 1 Introduction

Internal tides (IT) are internal waves generated by interactions between the barotropic tide and topographic features, such as ridges or continental slopes, and that propagate mainly at tidal frequencies. They are ubiquitous in the ocean, and play a crucial role in vertical mixing and energy transport, especially in the deep ocean (Munk and Wunsch, 1998; Vic et al., 2019). Propagating over large distances, they encounter regions with energetic mesoscale turbulence, and lose their fixed phase relationship with the astronomical forcing through the nonlinear interactions with this turbulence. The resulting incoherent internal tide field (often called "non phase-locked" or "non-stationary" internal tide in the literature), highly unpredictable, complicates for example our ability to disentangle internal tides and low-frequency turbulent signals from satellite data (Richman et al., 2012). Nelson et al. (2019); Zaron (2017) show that for the principal semidiurnal tide (M2) the incoherent part accounts for approximately half of the IT variance (44% in Zaron (2017)), averaged over the global ocean. There is therefore a need to better



understand how incoherent waves propagate, and to develop algorithms that could estimate and separate the surface signatures of the two dynamics in observational data.

In the oceanographic community, various studies examined the impact of a balanced (turbulent) jet on the inertia-gravity wave field properties in both realistic and idealised models. For instance, Ponte and Klein (2015) demonstrated on realistic simulations that the increase of energy of a fully turbulent jet contribute significantly to the loss of coherence of the wave

field. Kelly and Lermusiaux (2016); Duda et al. (2018) used geometric wave theory to examine internal tide reflections and refractions by the Gulf Stream in both realistic and idealised simulations, assessing the influence of the IT's propagation angle with the jet on waves deflections. Savva and Vanneste (2018) focused on the statistical properties of an internal tide scattered by a random flow from idealised simulation, where a kinetic transport equation is derived to study the wave energy propagation through a random media. In the range of modal decomposition technique, Dunphy et al. (2017); Kelly et al. (2016); Rainville

and Pinkel (2006) and more other studied the feasibility to study the propagation of IT on reduced order models based on vertical mode projection, adapted to express the vertical structure of IT. Ward and Dewar (2010) studied the scattering of inertia-gravity waves (IGW) by decomposing the non-linear interactions terms of a rotating shallow water (RSW) model on a normal mode decomposition, and using the theory of resonant triad. They showed that waves that resonantly interact with the jet transfer their energy to waves of equal wavelength, and that this transfer is intensified for short-wavelength and strong

background flows.

In this paper, we consider a data-driven approach to extract and analyse the wave field scattered by a balanced turbulent jet in a numerical simulation. Amongst the numerous data-driven decomposition techniques in fluid mechanics to extract meaningful physical quantities, we focus on methods based on the proper orthogonal decomposition (POD) (Berkooz et al., 2003; Lumley, 1967). The POD (often called empirical orthogonal functions – EOF – in the geophysical fluid dynamics community) is a

modal decomposition method to extract recurrent phenomena, namely coherent structures, in flow data (Long et al., 2021). Its frequency-domain variant, the spectral proper orthogonal decomposition (Towne et al., 2018; Schmidt and Colonius, 2020), provides a decomposition which is optimal to capture the variance associated with realisations of the Fourier spectrum. The method we propose, to examine wave scattering by turbulence, is based on the analysis of the complex wave amplitudes, extracted by complex demodulation, which is slowly varying (compared to the typical wave period) due to interactions with

a fluctuating jet. We investigate the scattering by extracting the correlations between the slow modulations of the wave and the coherent structure of the jet by applying the extended POD method (EPOD; Boree, 2003). Prior to this, we introduce the broadband POD method (BBPOD) to extract the most energetic modes of the broadband wave field, allowing for a comparison of the two decomposition methods. As it will be described, BBPOD can be viewed as a reformulation of SPOD, introducing explicitly the filtering frequency band, allowing us to interpret correctly the modes associated with the scattered field, due

to spectral broadening. For the sake of clarity in what follows, we will avoid using the term "coherent structures" of the (incoherent) wave field, and prefer the term optimal modes or simply modes of the wave, as the term "incoherent" have a different meaning here.

Based on the proposed decomposition method, we also address the issue of the estimation and separation of the internal tide and the mesoscale flow from observations. The development of algorithms has also recently received a great attention in the



literature. Some methods are based on physical approaches, such as in Ponte et al. (2017), who assumed a weak signature of IT on surface density fields, and consider potential vorticity dynamics to identify the balanced motion (BM) from the observations. As for more complex data-assimilation techniques, Le Guillou et al. (2021) developed a coupled iterative approach based on 4D-Var assimilation for the IT and back-and-forth nudging for the BM. In data-driven methods, there is a recent focus on deep-learning approaches to disentangle low and high frequency signal (Wang et al., 2022; Gao et al., 2024). Gao et al. (2024)

address in particular the difficulty caused by the long revisit time of altimeters by developing a method based on single SSH snapshots. Egbert and Erofeeva (2021) developed a method based on a decomposition of IT onto spectral EOF, in order to map satellite observations to a realistic model, and estimate the sea surface height (SSH) contribution of the internal tide. Here, we propose a similar methodology to the previous two, which aims at performing an additional step, taking advantage of the EPOD technique to have a complete estimate of the jet and the wave, including estimates of velocities, from a single SSH

observation. In particular, we address configurations where the jet dominates the flow, in which case the weak internal wave signal can still be estimated by correlation. The algorithm is tested on an idealised simulation of the one layer RSW model representing such a configuration.

   The plan of this paper is as follows. We begin in Sect.2 by describing the model used to investigate the dynamics of a wave field interacting with a zonal turbulent flow. In Sect.3, the different methods are introduced, namely the BBPOD (Sect.3.1)

and the EPOD (Sect.3.2). The algorithm to disentangle observations is presented in Sect.3.3. Section 4 presents the numerical results computed from idealised simulations. The variability of the wave field is examined in Sect.4.2, by means of the modal decomposition techniques of this study. Thereafter, estimates of the two dynamics are given in Sect.4.3.

## 2 Non linear wave-flow interactions

This section begins by presenting the RSW model used to investigate incoherent waves interacting with a turbulent flow.

Thereafter, we derive a simple equation for the incoherent complex wave amplitudes and we introduce some motivations of the method that will be presented in Sect.3.

### 2.1 One layer rotating shallow water model

To examine the propagation of internal tides through a nearly-balanced jet, we study the solutions of the one layer rotating shallow water (RSW) model, which is minimal to represent wave-jet interactions (*e.g.* Vallis, 2006; Ward and Dewar, 2010).

The equations are non-dimensionalized as follows.

$$t^{\sharp} = f_0^{-1} t; \; (x^{\sharp}, y^{\sharp}) = (l\,x, l\,y); \; \boldsymbol{v}^{\sharp} = (u^{\sharp}, v^{\sharp}) = (U\,u, U\,v); \; h^{\sharp} = H_o \frac{R_o}{B_u} h,$$

where the superscript $.^{\sharp}$ refers to the dimensional variables. The parameter $f_0$ is the Coriolis frequency, $\boldsymbol{v}$ is the horizontal velocity and $h$ is the SSH, with a layer thickness at rest $H_o$. The dimensionless parameters are the Rossby number $R_o = U/f_0 l$ and the Burger number $B_u = R_d^2/l^2$ where $R_d = \frac{\sqrt{gH_0}}{f_0}$ is the Rossby deformation radius. The characteristic timescale is taken

here as the inertial time $T = f_0^{-1}$, adapted for studying internal wave propagation, as it is the lower frequency bound of





the internal wave spectrum. The reference length scale $l$ and the reference speed $U$ are chosen to be of the order of the jet thickness and speed, respectively. Moreover, the Coriolis frequency follows the beta-plane approximation, which writes in its dimensionless form $f(y) = 1 + \beta y$, where $\beta = R_d/(R_T\sqrt{B_u})$, with $R_T$ the Earth radius. Including the $\beta$-term is necessary to obtain a zonal jet and to study the scattering of the wave in a small domain (Vallis, 2006). An wave forcing term $\boldsymbol{q}_{frc,\omega} = (h_{frc,\omega}, \boldsymbol{v}_{frc,\omega})^T$, which is kept under a general form in this section, is also added on the right hand-side of the equations.

Introducing this non-dimensionalisation into the RSW model leads to the following set of equations, defined on $\Omega \times \mathbb{R}_+$ with $\Omega \subset \mathbb{R}^2$ bounded:

$$\partial_t h + B_u \mathrm{div}(\boldsymbol{v}) = -R_o\big[(\boldsymbol{v} \cdot \nabla)h + h\,\mathrm{div}(\boldsymbol{v})\big] + h_{frc,\omega} \tag{1a}$$

$$\partial_t \boldsymbol{v} + (1 + \beta y)\boldsymbol{v}^\perp + \nabla h = -R_o(\boldsymbol{v} \cdot \nabla)\boldsymbol{v} + \boldsymbol{v}_{frc,\omega} \tag{1b}$$

where $\boldsymbol{v}^\perp = (-v, u)^T$.

## 2.2 Complex wave amplitude

The inertia-gravity wave is represented here with a slowly time-varying wave ansatz, which is adequate for representing an IGW propagating through the slowly varying jet responsible of the coherence loss. This assumption is borrowed from high-frequency asymptotic approximations methods, such as ray tracing or WKB methods (*c.f.* Bühler, 2014). Here, only a time scale separation between the jet and the wave is assumed in order to extract the two components by filtering. The total wave field $\boldsymbol{q}_\omega = (u_\omega, v_\omega, h_\omega)^T$ is thus expressed as:

$$\boldsymbol{q}_\omega(t) = \Re(\tilde{\boldsymbol{q}}_\omega(t)e^{i\omega t}), \tag{2}$$

where $\tilde{\boldsymbol{q}}_\omega(t)$ is the slowly varying complex wave amplitude vector. These complex wave amplitudes can be extracted from the data $\boldsymbol{q} = (u, v, h)^T$ by complex demodulation:

$$\tilde{\boldsymbol{q}}_\omega = \langle 2\boldsymbol{q}e^{-i\omega t}\rangle, \tag{3}$$

where $<\cdot>$ is a low-pass filter (in time) selecting the potentially broaden spectrum of the wave. This complex wave amplitude can be further decomposed into a coherent component $\mathbb{E}[\tilde{\boldsymbol{q}}_\omega]$, where $\mathbb{E}$ is the expectation operator, and the fluctuations which correspond to the incoherent component of the wave field.

## 2.3 Non-linear interactions terms

We propose in this section a simple investigation of the non-linear interaction terms leading to the ansatz of Eq. (2), highlighting how these terms act to produce the incoherent wave field. To this aim, we perform a linearisation of equations Eq. (1) assuming that waves are of small amplitude. For more generality, a similar development can be performed by only assuming a time scale separation between the two dynamics and a small Froude number $F_r = U_\omega/\sqrt{B_u} \ll 1$ for the wave, where $U_\omega$ is the typical amplitude of the wave velocity $\boldsymbol{u}_\omega$ (see Wagner, 2016). For the present discussion, however, a standard linearisation appears to



be sufficient to link the complex demodulation procedure and the role of quadratic non-linearities in the scattering process. We are looking for solutions of the RSW Eqs. (1) of the form $q = q_{jet} + \epsilon q_\omega$, where $q_{jet}$ is the slow jet current component, $q_\omega$ are the perturbative waves, and $\epsilon$ is the small parameter of the perturbation expansion. The forcing wave term is also at order $\epsilon$ and we consider it time-harmonic of the form $q_{frc,\omega} = \Re(\tilde{q}_{frc,\omega}(x,y)e^{i\omega t})$.

    Order 0 approximation leads to the following equation for the slow dynamics (which is not the focus of the study):

$$\partial_t h_{jet} + B_u \text{div}(\boldsymbol{v}_{jet}) = -R_o\left[(\boldsymbol{v}_{jet} \cdot \nabla)h_{jet} + h_{jet}\,\text{div}(\boldsymbol{v}_{jet})\right] \tag{4a}$$

$$\partial_t \boldsymbol{v}_{jet} + (1 + \beta y)\boldsymbol{v}_{jet}^\perp + \nabla h_{jet} = -R_o(\boldsymbol{v}_{jet} \cdot \nabla)\boldsymbol{v}_{jet}, \tag{4b}$$

and a wave equation is obtained at order 1:

$$\partial_t h_\omega + B_u \text{div}(\boldsymbol{v}_\omega) = -R_o\left[(\boldsymbol{v}_{jet} \cdot \nabla)h_\omega + (\boldsymbol{v}_\omega \cdot \nabla)h_{jet} + h_{jet}\,\text{div}(\boldsymbol{v}_\omega) + h_\omega\,\text{div}(\boldsymbol{v}_{jet})\right] + h_{frc,\omega} \tag{5a}$$

$$\partial_t \boldsymbol{v}_\omega + (1 + \beta y)\boldsymbol{v}_\omega^\perp + \nabla h_\omega = -R_o\left[(\boldsymbol{v}_{jet} \cdot \nabla)\boldsymbol{v}_\omega + (\boldsymbol{v}_\omega \cdot \nabla)\boldsymbol{v}_{jet}\right] + \boldsymbol{v}_{frc,\omega}. \tag{5b}$$

Let us now derive a simple equation for the incoherent complex wave amplitudes. For conciseness, we first write Eq. (5) in terms of linear and bilinear operators:

$$\partial_t \boldsymbol{q}_\omega + \boldsymbol{L}(\boldsymbol{q}_\omega) = -R_o\boldsymbol{B}(\boldsymbol{q}_{jet}, \boldsymbol{q}_\omega) + \boldsymbol{q}_{frc,\omega}, \tag{6}$$

with,

$$\boldsymbol{L}(\boldsymbol{q}_\omega) = \begin{pmatrix} B_u\text{div}(\boldsymbol{v}_\omega) \\ (1 + \beta y)\boldsymbol{v}_\omega^\perp + \nabla h_\omega \end{pmatrix}, \; \boldsymbol{B}(\boldsymbol{q}_{jet}, \boldsymbol{q}_\omega) = \begin{pmatrix} (\boldsymbol{v}_{jet} \cdot \nabla)h_\omega + (\boldsymbol{v}_\omega \cdot \nabla)h_{jet} + h_{jet}\,\text{div}(\boldsymbol{v}_\omega) + h_\omega\,\text{div}(\boldsymbol{v}_{jet}) \\ (\boldsymbol{v}_{jet} \cdot \nabla)\boldsymbol{v}_\omega + (\boldsymbol{v}_\omega \cdot \nabla)\boldsymbol{v}_{jet} \end{pmatrix}.$$

Here, $\boldsymbol{L}$ is a linear operator and $\boldsymbol{B}$ is a symmetric bilinear map associated to the non-linear interactions with the jet. From this equation, we can readily write an equation for the wave amplitude by injecting the wave ansatz (2) into (6) and multiplying the result by $e^{-i\omega t}$, and low pass filtering, thus obtaining:

$$\partial_t \tilde{\boldsymbol{q}}_\omega + i\omega\tilde{\boldsymbol{q}}_\omega + \boldsymbol{L}(\tilde{\boldsymbol{q}}_\omega) = -R_o\boldsymbol{B}(\boldsymbol{q}_{jet}, \tilde{\boldsymbol{q}}_\omega) + \tilde{\boldsymbol{q}}_{frc,\omega}. \tag{7}$$

    Now decomposing the jet component into a mean part $\mathbb{E}[\boldsymbol{q}_{jet}]$, where $\mathbb{E}$ is an expectation operator, and a fluctuating part
$\boldsymbol{q}'_{jet} = \boldsymbol{q}_{jet} - \mathbb{E}[\boldsymbol{q}_{jet}]$ gives the following equation:

$$\partial_t \tilde{\boldsymbol{q}}_\omega + i\omega\tilde{\boldsymbol{q}}_\omega + \boldsymbol{L}(\tilde{\boldsymbol{q}}_\omega) + R_o\boldsymbol{B}(\mathbb{E}[\boldsymbol{q}_{jet}], \tilde{\boldsymbol{q}}_\omega) = -R_o\boldsymbol{B}(\boldsymbol{q}'_{jet}, \tilde{\boldsymbol{q}}_\omega) + \tilde{\boldsymbol{q}}_{frc,\omega}. \tag{8}$$

We now make the assumption that the wave amplitudes evolves on slow time scales compared to the wave period $2\pi/\omega$ (which can be formalized more rigorously by a WKBJ approach). This allows us to write:

$$\tilde{\boldsymbol{q}}_\omega = -R_o\boldsymbol{R}\boldsymbol{B}(\boldsymbol{q}'_{jet}, \tilde{\boldsymbol{q}}_\omega) + \boldsymbol{R}\tilde{\boldsymbol{q}}_{frc,\omega}, \tag{9}$$





where $\boldsymbol{R} = [i\omega I + \boldsymbol{L} + \boldsymbol{B}(\mathbb{E}[\boldsymbol{q}_{jet}], \cdot)]^{-1}$ is the resolvent operator, with $i\omega$ assumed not to be an eigenvalue of $\boldsymbol{L} + \boldsymbol{B}(\mathbb{E}[\boldsymbol{q}_{jet}], \cdot)$. Thus, under the hyptoheses of small wave amplitude (or Froude number) and timescale separation between the wave period and the jet fluctuations, we find that the evolution of the complex amplitudes reflects instantaneously the jet fluctuations through the nonlinear interaction term and the resolvent operator. It can be noted that this time scale separation can also be interpreted by the fact that the resolvent operator is approximately constant over the frequency band considered for the broadband scattered

wave.

Separating the wave solution into a coherent and incoherent contribution $\tilde{\boldsymbol{q}}_\omega = \mathbb{E}[\tilde{\boldsymbol{q}}_\omega] + \tilde{\boldsymbol{q}}'_\omega$ and taking the expectation operator of Eq. (9) leads to an equation for the coherent wave amplitude:

$$\mathbb{E}[\tilde{\boldsymbol{q}}_\omega] = \boldsymbol{R}\big[\tilde{\boldsymbol{q}}_{frc,\omega} - R_o\mathbb{E}[\boldsymbol{B}(\boldsymbol{q}'_{jet}, \tilde{\boldsymbol{q}}'_\omega)]\big]. \tag{10}$$

This means that the coherent wave amplitude is the sum of the linear response of the forcing $\tilde{\boldsymbol{q}}_{frc,\omega}$, and an averaged non-linear

correction involving the jet fluctuations and the incoherent part.

Finally, substracting this equation Eq. (10) to Eq. (9) leads to the equation for the incoherent wave amplitude:

$$\tilde{\boldsymbol{q}}'_\omega = \underbrace{R_o\boldsymbol{R}\big[\mathbb{E}[\boldsymbol{B}(\boldsymbol{q}'_{jet}, \tilde{\boldsymbol{q}}'_\omega)] - \boldsymbol{B}(\boldsymbol{q}'_{jet}, \tilde{\boldsymbol{q}}'_\omega)\big]}_{\text{Multiple scattering}} - \underbrace{R_o\boldsymbol{R}\boldsymbol{B}(\boldsymbol{q}'_{jet}, \mathbb{E}[\tilde{\boldsymbol{q}}_\omega])}_{\text{single scattering}}. \tag{11}$$

In this equation, the non-linear interactions are decomposed into a primary scattering term, which is the interaction between the coherent wave $\mathbb{E}[\tilde{\boldsymbol{q}}_\omega]$ and the jet fluctuations, and a multiple scattering term which is the interaction with the jet fluctuations and

the incoherent wave component. It can be noted that the single scattering term only involves the coherent wave part (in addition to the jet fluctuations): thus, the fluctuations of the wave field (*i.e.* the incoherent wave field) are driven by the fluctuations of the jet. This decomposition of the wave response to the nonlinear interactions with the fluctuating jet will be exploited in Sect. 3.2, in order to provide grounds for using the EPOD method to analyse and estimate the coupled wave/jet fluctuations.

## 3 Methods

This section details the data-driven methods applied to investigate the wave variability due to a scattering by a jet. We should stress that it is important to capture the broadband spectrum of the wave field in order to study its dynamics and the scattering by the turbulent low-frequency dynamics. Indeed, in time frequency-space, the operator $\boldsymbol{B}$ in Eqs. (6) to (11) is a convolution operator between the dynamics of the wave and the jet. Under its action, the spectrum of an incident harmonic wave at frequency $\omega$, as we consider in this study, represented by a Dirac in spectral space, is broadened by convolution with the slow dynamics.

The nearby frequencies around $\omega$ are then associated to the incoherent components. Authors in the oceanographic literature often design the broadband spectrum of IT as "cusps". This has been highlighted in particular in Colosi and Munk (2006); Zaron (2022) by investigating spectra of high-resolved time records of Sea Level Height, showing that the incoherent frequency were actually not as low as expected. The spectral broadening of a scattered field has also been studied in various fluid mechanics contexts such as in Campos (1978); Clair and Gabard (2016), for non-dispersive waves.



It is thus necessary to design an algorithm that captures all these effects of triadic interactions with the jet, which is one of the motivation of the BBPOD algorithm, proposed in Sect. 3.1 below, which precisely extracts the optimal modes in a energetic sense for large frequency bands dynamics. The description of the EPOD method follows Boree (2003), to extract correlation within the data, adapted here to the case of wave-flow interaction. Finally, a simple algorithm of estimation is presented in Sect. 3.3, where the modal decompositions that are presented define an observation operator to estimate the two dynamics

from a non-filtered observation.

### 3.1 The Broadband POD method

The algorithm that we call Broadband POD (BBPOD) consists of performing a POD on a complex demodulated (wave) field, in order to capture the most energetic modes of variability of this wave field from numerical data. This algorithm can be viewed as a spectral proper orthogonal decomposition (Towne et al., 2018; Schmidt and Colonius, 2020), but which is

adapted to optimally represent finite width frequency band dynamics. The connection between the two algorithm is detailed in appendix A. This algorithm is also similar to a POD applied to wavelet transformed data (in time), in the sense that the complex demodulation also provides the content at a given frequency $\omega$. A wavelet EOF analysis of ocean waves has been performed in particular in Wang et al. (2000); Pairaud and Auclair (2005) applied either to the vertical direction or in time.

We consider a set of data $\boldsymbol{q}$, assumed to be statistically stationary (at least first and second order), and containing a broadband

peak centered around a frequency $\omega$. The algorithm of BBPOD to extract the coherent structures evolving at frequency $\omega$ is made of the following steps. We first compute the complex demodulation of $\boldsymbol{q}$ at the frequency $\omega$, $\tilde{\boldsymbol{q}}_\omega$ (Eq. (3)), with the choice of an appropriate filter to capture the broadband structure of the data. We next compute the space auto-correlation tensor of the complex amplitudes:

$$\mathbf{C}(x,y,x',y') = \mathbb{E}[\tilde{\boldsymbol{q}}_\omega(x,y,t) \otimes \tilde{\boldsymbol{q}}_\omega^*(x',y',t)], \tag{12}$$

where $\otimes$ is the dyadic product, product of the components $(q_i q_j)_{i,j}$, and the superscript $\cdot^*$ denotes the transpose-conjugate operation. Here, the expectation operator $\mathbb{E}$ is computed as a mean over time under the ergodicity hypothesis.

Before solving the POD problem, we define an innerproduct representative of the quadratic energy $E$ of the model Eq. (1), encoded with a positive definite matrix $\mathbf{W}_E$:

$$\|\boldsymbol{q}\|_{\mathbf{W}_E}^2 = (\boldsymbol{q}, \mathbf{W}_E \boldsymbol{q})_{L^2(\Omega)} \tag{13a}$$

$$= \frac{1}{2} \int_\Omega (u^2 + v^2)\, \mathrm{d}x\mathrm{d}y + \frac{1}{2B_u} \int_\Omega h^2\, \mathrm{d}x\mathrm{d}y. \tag{13b}$$

The BBPOD modes $(\boldsymbol{\psi}_{n,\omega})_n$ are then defined as the solution of the Fredholm equation:

$$\int_\Omega \mathbf{C}(x,y,x',y')\mathbf{W}_E(x',y')\boldsymbol{\psi}_{n,\omega}(x',y')\, \mathrm{d}x'\mathrm{d}y' = \lambda_{n,\omega}\boldsymbol{\psi}_{n,\omega}(x,y), \tag{14}$$



with non-negative eigenvalues $\lambda_{n,\omega}$. The BBPOD modes form an orthonormal basis of square integrable functions (in space), respectively to the innerproduct defined in Eq (13). The complex demodulated field can be expressed by the decomposition:

$$\tilde{\boldsymbol{q}}_{\omega}(t) = \sum_{n=0}^{\infty} a_{n,\omega}(t)\boldsymbol{\psi}_{n,\omega}, \tag{15}$$

where $a_{n,\omega}(t) = \int_{\Omega} \boldsymbol{\psi}_{n,w}^{*}\mathbf{W}_{E}\tilde{\boldsymbol{q}}_{\omega}(t)dxdy$ is the n-th projection coefficient. The modes are also decorrelated from each other and are optimal to express the quadratic mean energy at frequency $\omega$, calculated as:

$$\mathbb{E}(\|\tilde{\boldsymbol{q}}_{\omega}\|_{\mathbf{W}_{E}}^{2}) = \sum_{n=0}^{\infty} \lambda_{n,\omega}. \tag{16}$$

An important simplification arises in the presence of an homogeneous direction, that is if the statistics do not depend on
a coordinate. For our idealised test case, where the $x$ direction is homogeneous, the BBPOD modes dependence along this direction reduces to Fourier modes of the form $e^{ik_x x}$, where $k_x$ are wavenumbers. Yet this property is not leveraged in this study, as in Sect.3.3 will be presented an algorithm to estimate the low and high frequency dynamics for the most general wave-flow configuration. The reader can refer to the literature for more details of the theoretical background of POD methods (Towne et al., 2018; Schmidt and Colonius, 2020; Berkooz et al., 2003).

Lastly, in the literature, POD usually applies to a zero-mean process, and the mean is subtracted beforehand from the data. However, if the mean field is solution of the Fredholm equation above (Eq. 14) the procedure remains unchanged if we consider the total field or only the fluctuations. For more generality, and because the mean quantities are relevant both to examine our problem of wave scattering and for the estimations in Sect.4.3, the total field is considered here.

## 3.2 The extended POD method

In its original form, the extended POD (EPOD) method (see Boree, 2003) extracts, from a target field representing any physical quantities (*e.g* pressure), the contribution correlated to a given POD basis computed from any other quantity (*e.g* velocity). It is adapted here to the analysis of a wave scattered by a turbulent background flow. Indeed, as stated previously (Eq. 11), the incoherent contribution of the wave field is generated by the jet through the bilinear form $\boldsymbol{B}(\boldsymbol{q}_{jet}, \tilde{\boldsymbol{q}}_{\omega})$. The EPOD method can thus be leveraged to exploit this correlation to estimate the fluctuation of the wave field knowing the fluctuations of the jet, or
conversely. Here, we propose to extract the part of the wave correlated to the jet component, by the computation of its EPOD modes, forming a decomposition of the wave correlated to a POD decomposition of the jet component. The jet component is extracted from data by a low-pass filter.

The $n$-th EPOD mode is defined by:

$$\boldsymbol{\chi}_n(\tilde{\boldsymbol{q}}_{\omega}, \boldsymbol{q}_{jet}) = \frac{\mathbb{E}[\tilde{\boldsymbol{q}}_{\omega}\, a_n]}{\lambda_n}, \tag{17}$$

where $a_n$ is the projection coefficient of the jet component onto its POD basis (for the innerproduct defined in Eq. (13)) and $\lambda_n$ is the associated eigenvalue. As the POD modes of the jet are decorrelated between each other, the $n$-th EPOD contribution $a_n\boldsymbol{\chi}_n(\tilde{\boldsymbol{q}}_{\omega}, \boldsymbol{q}_{jet})$ is the part of the wave correlated to the $n$-th POD mode of the jet, but completely decorrelated to the other





modes of the jet. It provides a decomposition of the wave component $\tilde{q}_{c,\omega}$ that is correlated with the N-order truncation POD decomposition of the turbulent jet:

$$\tilde{q}_{c,\omega} = \sum_{n=0}^{N} a_n \chi_n(\tilde{q}_\omega, q_{jet}). \tag{18}$$

It can be noted that this procedure filter out all wave contributions decorrelated from the jet (incoming from outside the domain for instance). In order to lighten the notation in the following, we will drop the second argument $q_{jet}$ and denote by $\chi_n(\tilde{q}_\omega)$ the $n$-th EPOD mode.

It can be shown that the EPOD modes of the wave field and the POD modes of the jet are dynamically linked with each other through the resolvent operator. This confers a strong interpretation of this correlation-based technique, since it highlights an underlying connection between these modes through the dynamical model. To this aim, let us consider the Eq. (11) for the incoherent wave, in a regime where the multiple scattering terms can be neglected (*e.g.* if the incoming wave is coherent and the interaction zone remains of limited extent). Taking the expectation of this equation multiplied by a POD coefficient of the fluctuations $a'_n$ associated with mode $\psi'_n = (\psi'_u, \psi'_v, \psi'_h)^T$, one obtains the following equation:

$$\frac{\mathbb{E}[a'_n \tilde{q}'_\omega]}{\lambda'_n} = -R_o \boldsymbol{RB}\left(\frac{\mathbb{E}[a'_n q'_{jet}]}{\lambda'_n}, \mathbb{E}(\tilde{q}_\omega)\right),$$

that is:

$$\chi_n(q'_\omega) = -R_o \boldsymbol{RB}(\psi'_n, \mathbb{E}(\tilde{q}_\omega)). \tag{19}$$

This equation indicates that the wave EPOD modes are instantaneous responses to the interactions between the coherent wave and the jet modes, under the hypotheses of a flow dominated by single scattering interactions. This dynamical link between POD and EPOD has also been exploited in the context of wall-bounded turbulent flows (Karban et al., 2022) and turbulent jet flows (Karban et al., 2023).

As a final remark, we shall stress that EPOD modes are optimal for an appropriate innerproduct and vector field. Indeed, the vector $(\psi(q_{jet}), \chi(\tilde{q}_\omega))^T$ is an approximate solution of the POD problem associated to the "extended" vector $(q_{jet}, \tilde{q}_\omega)^T$, with the associated weight matrix $\mathbf{W} = (\mathbf{W}_E, \epsilon \mathbf{W}_E)$, when $\epsilon$ goes to 0. For this procedure, POD maximises the energy of the jet contribution, while the wave components are only correlated to the jet field. In the assumption of small wave amplitude, the total energy is essentially the one of the jet, and we could compute almost identically a POD with weight $\mathbf{W} = (\mathbf{W}_E, \mathbf{W}_E)$, such that $(\psi(q_{jet}), \chi(\tilde{q}_\omega))^T$ are the optimal modes of the full dynamics. The small wave amplitude is precisely the hypothesis performed in Sect. 2.3. This remark motivates using such a basis to perform a full reconstruction of the two motions for a dominant jet, as we propose in the next section.

### 3.3 Estimation algorithm

We now present a simple method to estimate IGW and BM from SSH observation in which both signals are entangled, in configurations where this observation is a single snapshot in time with partial coverage in space.



The observation $\mathcal{Y}$ writes as $\mathcal{Y}(t) = h_{obs}(t) = \mathbb{1}_{\Omega_{obs}} h(t)$, where $\Omega_{obs}$ is the mask applied to the observations. The algorithm is as follows:

---

**Algorithm 1** Estimation

---

1. **Training stage**: Build the observation operator offline, where the modal set of the wave and the jet are computed. The operator is simply representing the superimposition of the jet and the wave component.

    (a) Compute the POD modes for the balanced dynamic $\psi_n(\boldsymbol{q}_{jet})$.

    (b) Compute the complex demodulation of the SSH of the wave, and the associated EPOD modes $\chi_n(\tilde{h}_\omega)$ (with an implicit dependence on $\boldsymbol{q}_{jet}$).

    (c) Build the observation operator $\mathbb{H}$ associated with the N-mode truncation of the SSH field:

$$\mathbb{H}(\boldsymbol{q}(t)) = \left(\hat{h}_{jet} + \hat{h}_\omega\right) \mathbb{1}_{\{(x,y)\in\Omega_{obs}\}} \tag{20}$$

$$= \Re\Big[ \sum_{n=0}^{N} a_n(t)\big(\psi_n(\boldsymbol{q}_{jet})_h + \chi_n(\tilde{h}_\omega)e^{i\omega t}\big)\Big] \mathbb{1}_{\{(x,y)\in\Omega_{obs}\}}, \tag{21}$$

where the jet component is approximated by its POD decomposition truncated at mode N:

$$\hat{h}_{jet} = \sum_{n=1}^{N} a_n(t)\psi_n(\boldsymbol{q}_{jet})_h$$

and the wave is expressed by its correlation to the jet (following Eq. (18)) and accounting for the fast wave complex exponential:

$$\hat{h}_\omega = \Re\big[\tilde{h}_{c,\omega}e^{i\omega t}\big] = \Re\Big[\sum_{n=0}^{N} a_n(t)\chi_n(\tilde{h}_\omega)e^{i\omega t}\Big].$$

where the ˆnotation refers to estimated fields.

2. **Estimation stage**:

    (a) For each snapshot, compute the coefficients which minimise the error with the observation:

$$\boldsymbol{a}^{LSQ}(t) = \min_{(a_0(t),\cdots,a_N(t))} \|\mathbb{H}(\boldsymbol{q}(t)) - \mathcal{Y}(t)\|^2_{L^2(\Omega_{obs})}. \tag{22}$$

    (b) Estimate the complex wave amplitude (following Eq. (18) ) and jet vectors:

$$\hat{\tilde{\boldsymbol{q}}}_\omega(t) = \sum_{n=0}^{N} a_n^{LSQ}(t)\chi_n(\tilde{\boldsymbol{q}}_\omega), \tag{23a}$$

$$\hat{\boldsymbol{q}}_{jet}(t) = \sum_{n=0}^{N} a_n^{LSQ}(t)\psi_n(\boldsymbol{q}_{jet}). \tag{23b}$$

An estimate of the true wave field can be recovered with the fast wave part $e^{i\omega t}$ following Eq. (2).

---

As mentioned previously, through the EPOD method, the estimated wave is assumed to be completely correlated to the jet via this algorithm. We take advantage of the presumed strong correlation between the slow BM and the complex incoherent





wave amplitude to estimate these two motions, and thereby could enable estimating the IGW field in configurations where the corresponding signal is dominated by a strong background flow. As suggested in Zaron (2017), this configuration is challenging for the estimation of the wave from real altimetric observations, which is the case for instance in the Gulf Stream region.

However, the part of the wave that is completely decorrelated from the jet is not estimated in this algorithm, as it is a quantity that is more difficult to quantify. In addition, velocity fields for both motions can be estimated from SSH by correlation, which thus provides an alternative method to using the geostrophic balance for BM and polarisation relations for IGW. The present algorithm is also minimal and does not require any temporal sampling of the observations. Only a single snapshot is considered to estimate the IGW fields and BM. We also did not made any assumption on the recovering of the coherent wave, which is still

a difficult task in regions with strong mesoscale flows (Ubelmann et al., 2022). Yet, if assuming that the coherent wave can be well estimated from observations, it could be substracted from the dataset in the training stage as well as from the observations (estimation stage). Besides, it can be noticed that no regularisation is considered in (22). We rely indeed on a "rigid" structure of the problem conferred by the modal decomposition, and then a small ratio between number of parameters and size of the observation space. This point will be discussed in the results Sect.4, and regularisation term could be added without loss of

generality.

IGW and BM estimates will be shown for an idealised simulation in Sect.4.3, but representative of the configuration of very small amplitude waves. To our knowledge, few studies have addressed the problem of the extraction of an inertia-gravity wave propagating through a strong background flow.

## 4 Numerical results

In this section, we present some results on the ability of the previously introduced methods to extract and estimate the IGW and balanced flow field in numerical simulations of the RSW Eqs. (1). The first part of the results (Sect. 4.2) concerns the study of the variability of the IGW field in relation to the turbulent jet fluctuations. We next focus on the estimates of the IGW and BM fields (Sect. 4.3).

### 4.1 Numerical configuration

Five numerical simulations of Eqs. (1) featuring a plane wave interacting with a zonal jet have been performed. The parameters that vary from one simulation to another are the temporal frequency of the incoming wave, its direction of propagation, and the Rossby number of the turbulent jet – see Table 1. The simulations are labelled W1, which is the reference simulation, then W2 to W5. The Burger number is $B_u = 1$ for all simulations (units for space coordinates is therefore the Rossby radius of deformation).

The equations have been discretised using a spectral method in space and a Runge-Kutta time scheme, using the open-source code Dedalus (Burns et al., 2020). The domain $\Omega$ is a doubly periodic rectangular domain of size $[0, 20] \times [-20, 20]$, discretised on a $128 \times 256$ grid for W1-W4, and on a $256 \times 1024$ grid for W5 which requires a higher resolution. All simulations are initialized with an eastward zonal jet at geostrophic equilibrium with a small perturbation superimposed to trigger its




| Numerical simulations | | | | | |
|---|---|---|---|---|---|
| Parameters | W1 | W2 | W3 | W4 | W5 |
| Frequency $\omega$ | $2f_0$ | $3f_0$ | $2f_0$ | $2f_0$ | $2f_0$ |
| Mode number $m_x$ | 0 | 0 | 1 | -1 | 0 |
| Rossby number $R_o$ | 0.2 | 0.2 | 0.2 | 0.2 | 0.35 |

**Table 1.** Parameters of the different simulations.

destabilisation. During the experiment, an eastward wind forcing of the form of a Gaussian function in $y$ is applied to maintain
the balanced current in a stationary state. In order to obtain stationarity, since energy is constantly added by the wind, a radiative damping term is added on the continuity equation of the form $\alpha h$, where $\alpha$ is a constant parameter (following *e.g* Brunet and Vautard (1996)). To ensure numerical stability, a small hyperviscosity diffusion term is also added in the equations. Once stationarity is reached (after $4000f_0^{-1}$, which corresponds approximately to 450 days at mid-latitude), for each simulation a northward propagating plane wave is generated in a nudging layer in the south of the domain. The properties of the nudging
follows the dispersion relation and the polarisation relation computed from the harmonic solutions of Eqs. (1) linearised around a state at rest. The simulation continues, with snapshots stored every $1/10$ wave period, until the dataset $\boldsymbol{q} = (u,v,h)^T$ contains a sufficiently large number of realisations of the dynamics (corresponding to $4000f_0^{-1}$ for W1-W4 and approximately to $8000f_0^{-1}$ for W5). Indeed, like any statistical methods, and in particular for POD methods, the ensemble of realisations has to be sufficiently large so that the BBPOD or EPOD modes computed numerically are well converged. We refer the reader to
Schmidt and Colonius (2020) on convergence issues. Note also that a sufficient sampling in time is required for extracting the complex wave amplitudes by time-filtering in Eq. (3).

The wave is extracted from the outputs of the simulation by complex demodulation around frequency $\omega$, and the jet by a low-pass filter. A fourth-order Butterworth filter is used, discarding frequency above $f_0/10$, corresponding approximately to periods less than 3 days. In Fig.1 is plotted the magnitude spectrum of the SSH field showing the broadband spectral peak
associated to the jet around $\omega = 0$ and the wave around $\omega = 2$, highlighting the time scale separation. We notice also the weak super-harmonic signal of IGW (at $\omega = 4$), which we do not address in this study. These simulations are representative of configurations where the wave has weak amplitude compared to the low-frequency dynamics, as described in Sect. 2.3. For the W1 run for instance, the SSH contribution of the wave represents only $1-2\%$ of the one of the background flow. For the estimations that are presented in Sect.4.3, even though the simulations are highly idealised, this is a challenging case to
estimate the wave from one entangled observation, due to its very small signal to noise ratio in the observation. This scale separation in amplitude is for example typical of the values that are found in the Gulf Stream region (*c.f.* Richman et al., 2012).

An example of snapshot associated with the reference simulation is shown in Fig.2.

In the following, the BBPOD modes for the wave will be denoted $(\lambda_{n,\omega}, \boldsymbol{\psi}_{n,\omega})$, while the POD modes of the jet are implicitly
denoted $(\lambda_n, \boldsymbol{\psi}_n)$. They are computed on the physical domain, after removing the sponge regions (for $|y| > 12$).



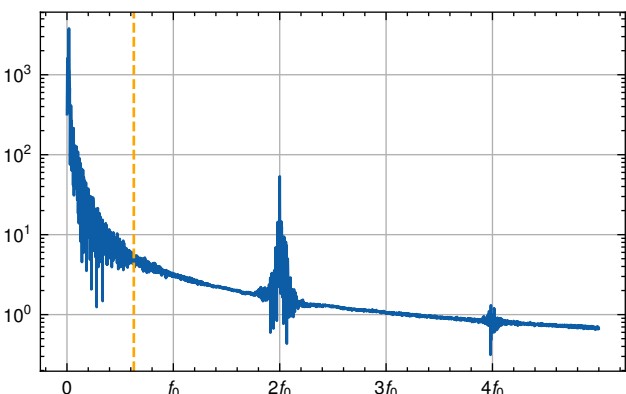

**Figure 1.** Magnitude spectrum of the SSH field in lin-log scale, in the center of the domain $x = 10, y = 0$, for W1.

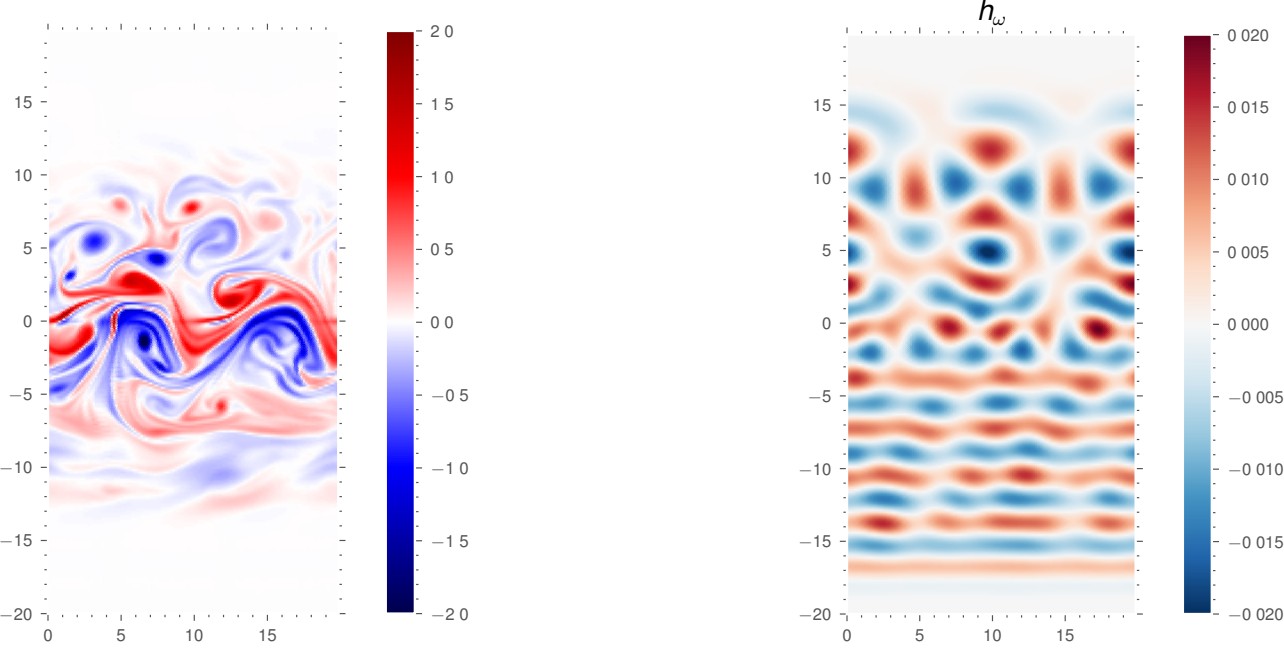

**Figure 2.** Snapshots of the reference simulation W1 with $\omega = 2f_0, m_x = 0$. In (**a**) is the total vorticity field and (**b**) shows the wave SSH contribution.



## 4.2 BBPOD and EPOD analysis of wave incoherence

In this section, we study the variability of the wave by means of the computation of its BBPOD and EPOD modes. We analyse the spatio-temporal features extracted from the different simulations, and we make a connection between the low and the high frequency dynamics to understand how the incoherent wave is generated by interaction with a turbulent jet in this context.

### 330 4.2.1 BBPOD analysis

In Fig. 3 are presented the SSH component of the first three weighted modes ($\sqrt{\lambda_{n,\omega}}\psi_{n,\omega}$) of the wave field propagating through the jet, for the different simulations described previously. The leading BBPOD mode, shown on the first column, is the most energetic one in the sense defined in Eq. (13). In Fig. 4 is plotted the time evolution of the associated leading BBPOD coefficients for W1, and the associated mean and root-mean square values. It shows that the first coefficient is nearly constant and equals its mean values, which indicates that this first mode is phase-locked with the nudging and thereby corresponds to the coherent part of the wave field, as commonly defined in the literature and discussed in the previous section 3. As we included in the computation of the energy the incident south part of the wave, which is essentially coherent, the whole coherent wave dominates energetically and stands out as the leading BBPOD mode. Consequently, higher modes are associated with the incoherent part of the wave field and have almost zero mean, as also indicated on Fig. 4. As visible in Fig. 3 (left column), the coherent mode exhibits monochromatic waves (in the x-direction) with the same propagation direction as the incident wave. We can distinguish a drop in amplitude in the north part of the domain, which is particularly pronounced for W3 (with $\omega = 3f_0$) and W5. This observation suggests stronger non-linearities, leading to a greater loss of energy to the incoherent waves in this region, which corroborates the analytical results of Ward and Dewar (2010) for small scale waves and strong background flows.

On the second and third column of Fig.3 are shown the two most energetic incoherent modes of each simulation. They are by definition uncorrelated to the first coherent mode. These modes are essentially non-zero in the north domain, and corresponds to nearly-plane waves that are deflected compared to the incident propagation direction with mode number $(-2,2),(-2,2),(3,-1),(-3,1)$ from top to bottom. Their structure will be investigated later in Fig. 6, by studying the correlation with the jet. Note that they do not represent perfect Fourier modes in the $x$ direction, and slights deviations to a single-mode structure may be due to statistical convergence effects.

Finally, we complete the description of the BBPOD modes by showing the modal energy distribution of the incoherent wave field, in Fig. 5, truncated up to 30 modes. The incoherent energy is approximately the same for waves propagating through the flow associated to $R_o = 0.2$, and as expected for the stronger jet the energy is significantly higher. Yet, each mode has a similar contribution on the total energy for each simulation, and the first 10 modes capture a large fraction of it, about $75 - 85\%$ of the incoherent part. The slightly less efficiency of the reconstruction for the fast wave $\omega = 3f_0$, and the strong jet, still indicates a more homogeneous modal distribution, consistent with a stronger impact of non-linearities. The graph in Fig.5 is equivalent to computing the mean-squared error (MSE) at mode N : $MSE = 1 - \sum_{n=1}^{N} \lambda_n / \sum_{n=1}^{+\infty} \lambda_n$.





**Figure 3.** From left to right the three dominant weighted BBPOD mode of SSH, calculated as $\sqrt{\lambda_{n,\omega}}\psi_{n,\omega}$. From first to last row are the waves W1 to W5.




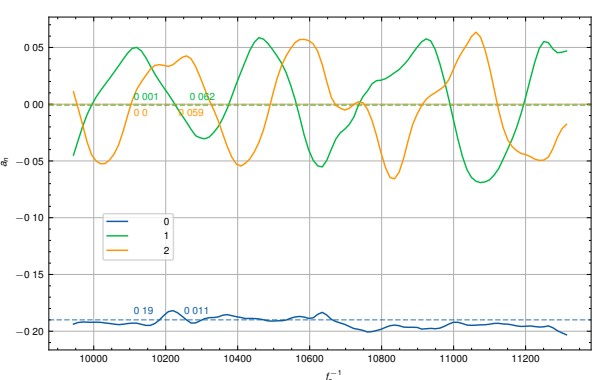
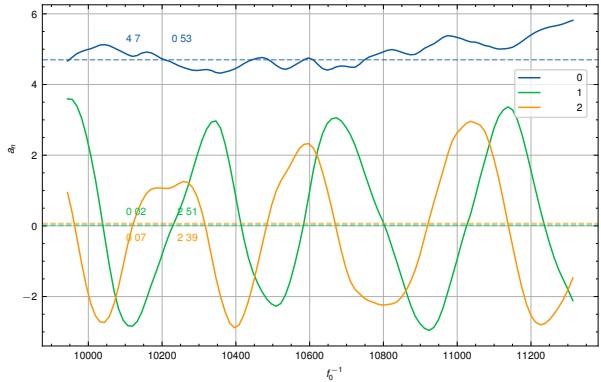

**Figure 4.** In (**a**) the three leading BBPOD coefficient of the wave $a_{n,\omega} = (\tilde{\boldsymbol{q}}_\omega, \boldsymbol{\psi}_{n,\omega})_{\mathbf{W}_E}$. In (**b**) the three leading POD coefficient of the jet: $a_n = (\boldsymbol{q}_{jet}, \boldsymbol{\psi}_n)$. They are computed from the reference run W1, and the respective mean values and root-mean squared errors are shown.

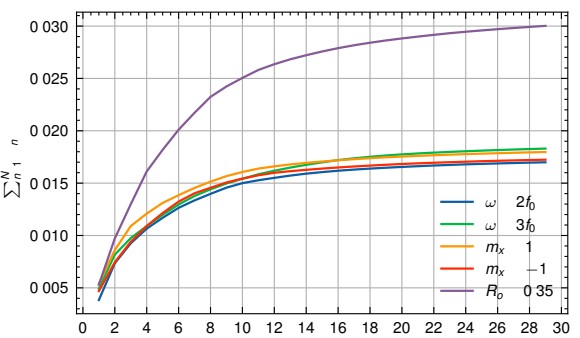
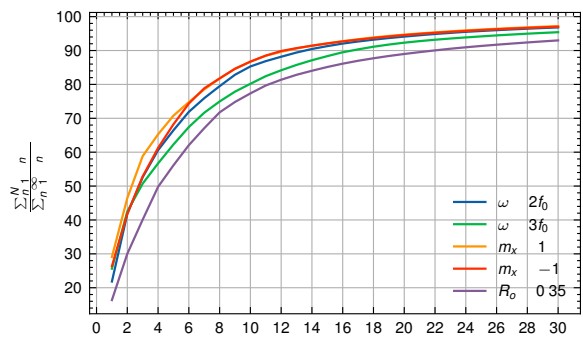

**Figure 5.** Modal energy distribution of the different incoherent wave fields. In (**a**) is the cumulative incoherent energy at mode N: $\sum_{n=1}^{N} \lambda_{n,\omega}$, and in (**b**) it is normalized by the total energy: $\sum_{n=1}^{N} \lambda_{n,\omega} / \sum_{n=1}^{+\infty} \lambda_{n,\omega}$.

### 4.2.2 EPOD analysis

Figure 6 shows the leading three POD modes of the jet, and the associated EPOD modes of the complex wave amplitudes

for the W1-4 runs. The most energetic mode of the jet is the mean field, and the two suboptimals shown are the dominant meandering structures of the jet, in phase-quadrature with each others. As for the wave, we show in Fig. 4 the three first POD coefficients. The leading coefficient is nearly constant in time, and is indeed associated to the mean flow, while the fluctuations are nearly zero mean.

The lower panel (**b**) in Fig.6 shows that the part of the wave correlated to the mean jet is the coherent wave, *i.e.* it is

very similar to the leading BBPOD mode of the wave field (left column in Fig. 3). This readily follows from the fact that both the first jet POD mode wave BBPOD mode essentially capture the mean flow. More interestingly, the second and third







**Figure 6.** Three leading POD modes of the jet (**a**) and the associated EPOD modes of the wave field (**b**) for simulations W1-W4. They are weighted by the square-root of the respective POD eigenvalue of the jet $\sqrt{\lambda_n}$.





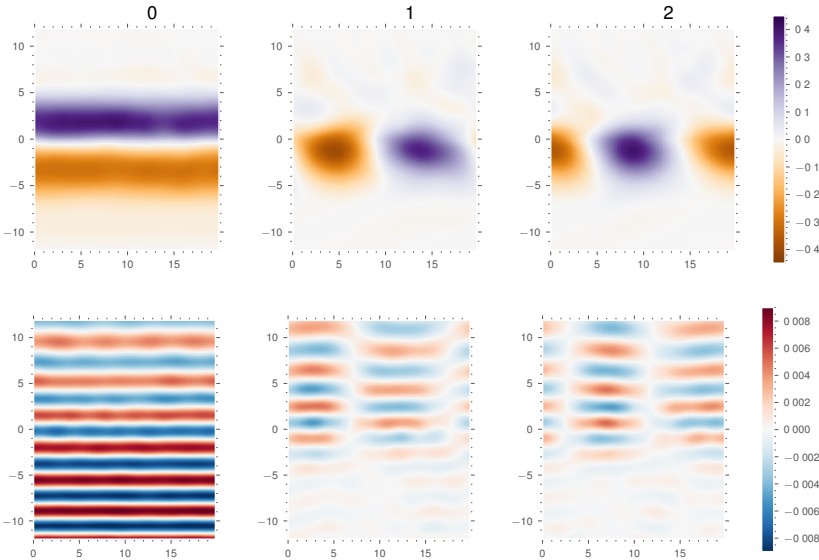

**Figure 7.** Three leading POD modes of the jet (**a**) and the associated EPOD modes of the wave field (**b**) for simulation W5. They are weighted by the square-root of the respective POD eigenvalue $\sqrt{\lambda_n}$.

EPOD modes, which are correlated to the meanders, exhibit stationary wave patterns, which are also in phase quadrature. It is especially remarkable for simulations with an incoming wave perpendicular to the jet (W1 and W2), where four nodes are visible along the $x$ direction. Similar patterns are found for the W5 run in Fig. 7. However, the meanders have here a twice
larger zonal wavelength and also have a weaker energetic contribution (see colorbars). The associated EPOD mode of the wave is a stationary wave with now 2 nodes in the domain. There is thus a clear correspondence between the wavelength of the modes of the jet and the wavelength of the stationary wave. We easily infer from Eq. (19), that these wave patterns result from the single interaction between each POD mode of the jet and the coherent wave. As stated theoretically before in Sect. 3.2, the EPOD modes computed from these numerical simulations are thus the instantaneous response of the interactions between the
jet and the coherent wave. All these configurations exhibit a dominance of the single scattering interactions by comparison with the multiple scattering terms. This probably owes to the fact that the dominant fluctuations patterns are localised in the center of the domain, and therefore the wave that is scattered upward cannot reinteract with these structures. We also infer that the present stationary waves result from the superposition of the two upward deflections extracted by the BBPOD method. In detail, in the BBPOD, the two deflections are decorrelated because associated with opposite zonal wavenumbers, which explains this
separation in two distinct modes, while they are recovered at once by correlation to the jet modes. Finally, by looking at the temporal coefficients of the two dynamics in Fig. 4, the time evolution of the wave modes seems to be completely driven by the time evolution of the jet modes, the two sharing the same time scale. Therefore, we can state that in these configurations the dominant structures of the jet completely drive the optimal modes of the wave, according to the aforementionned single triad interaction.





Note also the presence of a weak reflected signal by the meanders in the south part. The potential role of the spatial variability of the jet on IT, which includes the meanders, has also been suggested in Kelly and Lermusiaux (2016) by studying the reflections by the Gulf Stream.

To conclude about this investigation, the method extracts from the data an energetic incoherent wave of the form of a stationary wave, composed by deflections by the meanders. This wave accounts between $40 - 50\%$ of the incoherent energy

for W1-W4 and one third for W5 (see Fig. 5), and is the result of a triadic interaction between the coherent wave and the meanders. Even though the incoherent wave is not particularly weak, we find that, in the present configuration, single scattering interactions are still dominating the incoherent propagation of the wave through a jet with limited spatial extent. Furthermore, the correlation between the most energetic incoherence and the dominant structures of the balanced motion motivates the development of estimates for the full dynamics, with such modal decomposition, in case of entangled observations. This is

shown in the next section.

### 4.3 Estimation

The estimation technique (1) is performed for the reference simulation W1. We start in this study by analysing the capability of the technique to disentangle the two dynamics from a single observation. Then, we evaluate the sensitivity of the estimates to a spatial deterioration of the observations. In practice, we separate the time series into a training and a testing dataset. The

POD and EPOD modes are computed from the former, which has an approximate duration of $1300$ wave periods (16 months of simulation). The dataset for testing contains $50$ snapshots of the complex amplitudes separated from $4$ periods, so that the full time series contains $200$ wave periods.

#### 4.3.1 Estimation for a full SSH observation

Figure 8 shows estimates of the SSH contribution of IGW and the jet, from an observation of the total SSH, using 30 POD/E-

POD modes. It indicates an accurate estimate of the two dynamics, where the dominant patterns are recovered, with a small point-by-point error. For both, the error shows a disorganised field with no evidence of a well identified structure, which would not be captured by the method. To compare with a naive method, another estimate of the wave is shown in appendix. A (Fig. A1) computed with the optimal bases for the two dynamics (POD for the jet and BBPOD for the wave). While being optimal, the BBPOD basis cannot estimate the wave component at all, and requires a more sophisticated algorithm at least. It highlights

therefore the interest of the EPOD method in this configuration, for a simultaneous estimation of the jet and the wave.

In Fig. 9 is plotted the mean energy captured by an estimate of the wave taking 30 modes, at each point of the domain. The graph shows a quite accurate estimate of all the wave components, with more than $50\%$ of the energy recovered at approximately each point of the domain. It thus shows a simple data-driven alternative to recover the velocities component without the knowledge of the polarisation relation of the wave field. It suggests also that the estimated snapshot of Fig. 8 is

representative of the ability of the algorithm to disentangle the low and high frequency dynamics from the dataset of test.

Lastly, in Fig.10 is quantified the performance of the proposed method by computing the time-averaged $L^2$ norm error for the state vector $(h, u, v)^T$ associated with each type of motion, *i.e.* $\mathbb{E}\big[\|\hat{\tilde{\boldsymbol{q}}}_\omega - \tilde{\boldsymbol{q}}_\omega\|_{L^2}^2\big]/\mathbb{E}\big[\|\tilde{\boldsymbol{q}}_\omega\|_{L^2}^2\big]$ for the wave field and





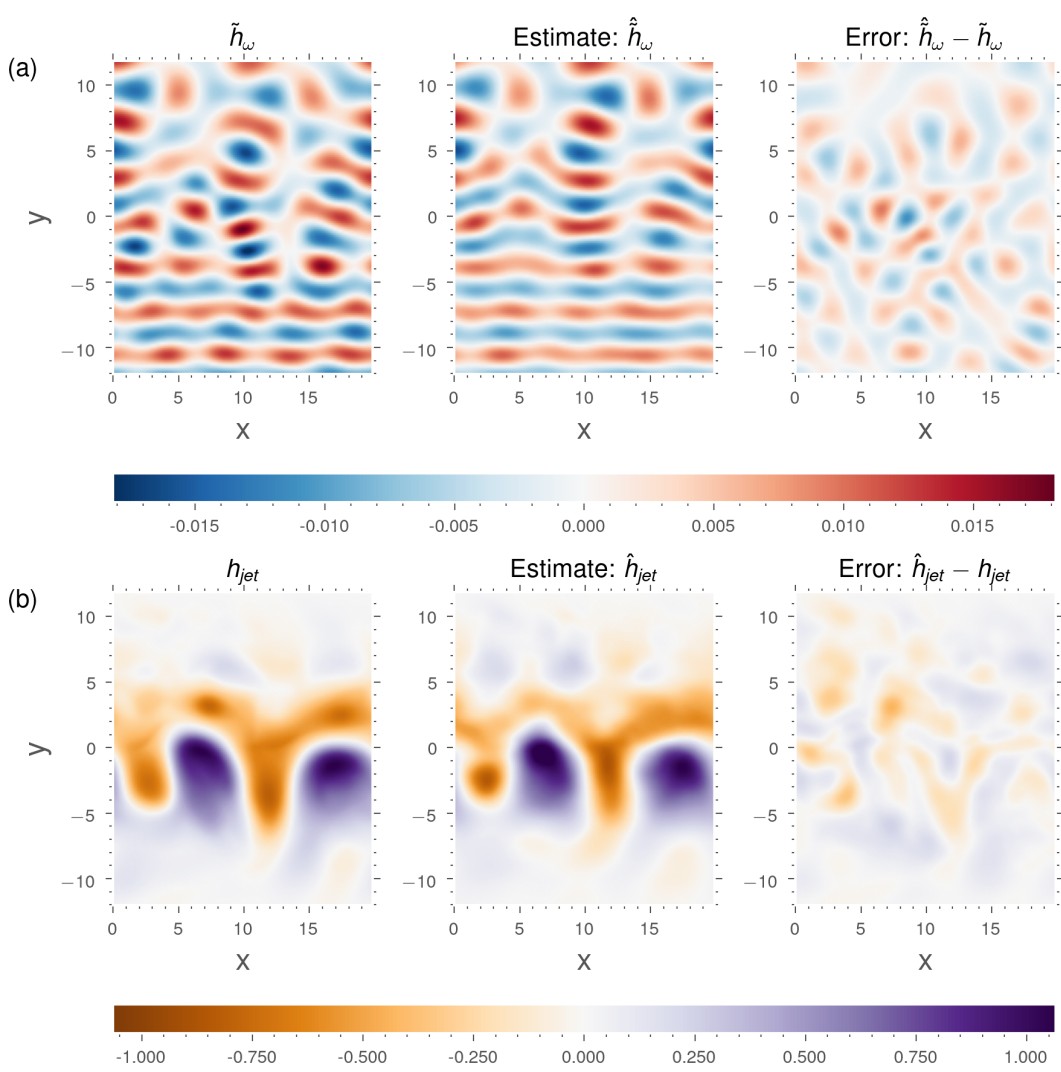

**Figure 8.** Estimate of the SSH contribution of the wave (**a**) and of the jet (**b**) for 30 modes from one snapshot, for W1.





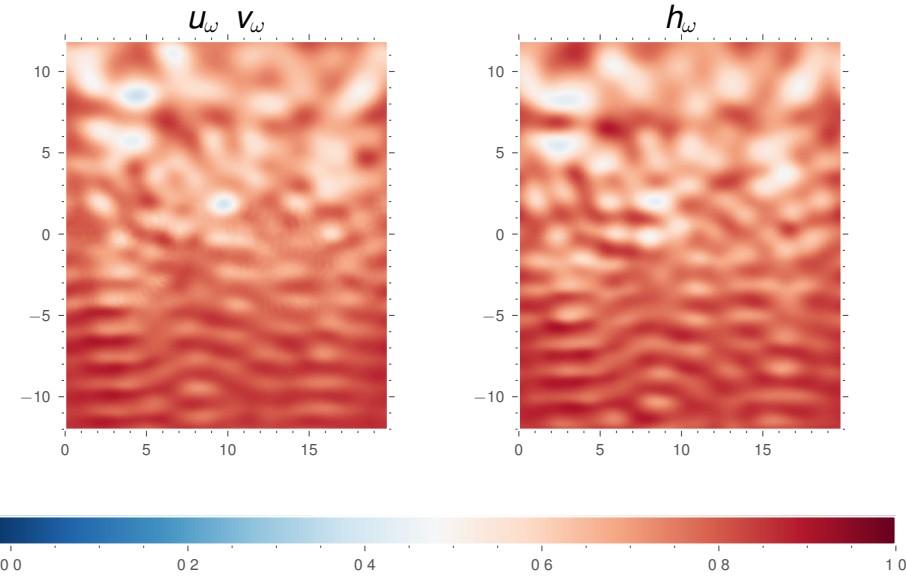

**Figure 9.** Normalized mean energy reduction of the wave field $(\tilde{u}_\omega, \tilde{v}_\omega, \tilde{h}_\omega)$ : $1 - \mathbb{E}\big[|\hat{\boldsymbol{q}}_\omega - \boldsymbol{q}_\omega|^2\big]/\mathbb{E}\big[|\boldsymbol{q}_\omega^2|\big]$, for an estimate over 30 modes, for W1.

$\mathbb{E}\big[\|\hat{\boldsymbol{q}}_{jet} - \boldsymbol{q}_{jet}\|_{L^2}^2\big]/\mathbb{E}\big[\|\boldsymbol{q}_{jet}\|_{L^2}^2\big]$ for the low-frequency motion. The estimate errors are compared with the averaged projection error performed when the fields are projected on their respective POD basis (POD for the jet and BBPOD for the wave). Thus,

for the wave the BBPOD projection corresponds to the projection of the wave onto its BBPOD modes following Eq. (15). By optimality of BBPOD and POD, the latter decompositions minimise the residual variance, hence it provides a lower bound for the error associated with the estimate. The figure indicates that the algorithm estimates properly all components of the jet field vector, while only $h$ is observed. For the wave part, more than $70\%$ of the total energy and $63\%$ of the incoherent energy is recovered. Both errors decrease with the number of modes up to 30 modes, suggesting that a large number of modes can be

taken for performing estimates. A slope break is visible in the errors at the third mode, which corresponds to the dominance of the meanders and the associated wave response, as discussed in Sect. 4.2. The energy distribution of higher modes is more homogeneous leading to a slower decreasing error. Note that there is a small amount of energy that is lost for the coherent part (see estimate at mode 1 in Fig. 10) between the estimate and the BBPOD projection. The EPOD projection error, which corresponds to the decomposition of the wave that is correlated to a N-th order decomposition of the jet is computed by

projecting the jet onto its POD coefficients (following Eq.(18)), is also shown in the figure (panel **a**). The small gap between the EPOD projection error and the estimate error (which turns out to be smaller) shows that this simple algorithm can accurately estimate the part of the wave that is correlated to the jet. Indeed, this stems from the fact that the jet coefficients are optimally recovered as stated before (see Fig. 10**b**). Finally, the gap between the estimate error and the BBPOD projection error associated with the wave field for a large number of modes shows some limitation of this method. Apart from the missing coherent signal

accounting approximately for $8\%$ of the total signal, this can be explained firstly by the fact that only the part of the wave





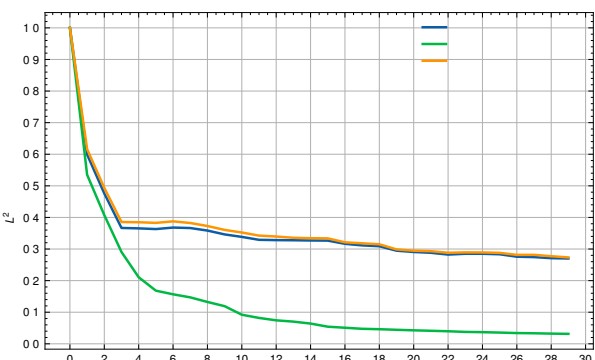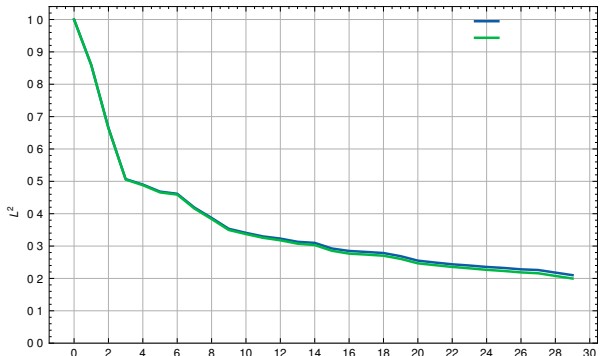

**Figure 10.** Time-averaged $L^2$ norm errors (blue) as a function of mode numbers included, together with the projection error of the corresponding POD decompositions, for the wave field (**a**) and the turbulent jet (**b**). For the wave, the EPOD decomposition Eq.(18) is also plotted (orange). It is performed for the W1 run.

correlated to the jet is estimated, as stated in Sect.3.3. Another reason is that the jet is only estimated at $80\%$ (with 30 modes) and therefore, the wave estimate misses the part correlated to the remaining $20\%$ of the jet. For a turbulent jet that can be expressed by fewer modes, that is said "low-rank", the algorithm is expected to perform slightly better at least. In appendix. A is shown in Table A1 the maximal values of averaged energy captured on the first 30 modes, for the simulations W1-W5, as it

shown in Fig.10. This highlights that apart from the small scale wave simulation, the algorithm can estimate more than $73\%$ of the total wave field, and between $44\% - 63\%$ of the incoherent part, even for the strong mesoscale flow associated with an energetic incoherent wave (Fig. 5). While reducing some of the variance of the incoherent part, the algorithm performs worse for the run W2 ($\omega = 3f_0$), associated to the fast wave, and starts increasing from the 15 modes (not shown). Yet, this frequency is superior to the $M_4$ frequency component, which is rather small in practice.

**4.3.2   Impact of a partial observation**

For the last part of the study, we intend to evaluate the sensitivity of the estimation algorithm to a degradation of the spatial sampling. The sparse observation generated consists of 4 vertical bands of SSH with a typical width of one Rossby radius (see Fig. 11) and homogeneously distributed in the domain. Combined with the fact that we do not have information in time, estimating and disentangling the two dynamics is a challenging task.

Figure 12 shows a snapshot of the estimated wave and jet fields, computed with 12 modes. The wave estimate exhibits a smoother field compared the the full observation case, as only 12 modes were taken to produce the estimate. The large scale dominant pattern are yet still captured, indicated by a small error in all the domain. The estimate for the jet captures the global shape, even if there is a more pronounced error compared to the full observation case, with some overestimated amplitudes.

    In Fig. 13 is shown the time-averaged $L^2$ norms errors for different spatial coverage of the domain, ranging from 10% (two

bands) to 20% (four bands). The results show that the algorithm is robust to capture the first three modes of the two dynamics





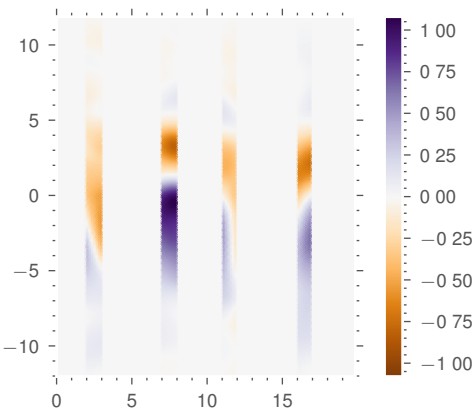

**Figure 11.** Sparse SSH snapshot observation from the W1 run, covering 20% of the full domain.

for a 20% coverage, and the error still decrease until 12 modes approximately. However, at 15% coverage and below, the convergence of the error is lost. In particular, we find that for very sparse observation coverage (10% of the domain) both estimates are diverging. In this case, the jet is not well estimated and, consequently, the wave cannot be accurately estimated only by correlation. A proper estimate of the balanced motion is central to accurately predict the wave, and this test case
illustrates one limitation of our algorithm. It can be noticed as well that high-order mode's estimation is more sensitive to this progressive observation degradation, suggesting that these high-order modes are more difficult to estimate and could lead more likely to a loss of robustness than a gain in accuracy.

Mitigation of this issue could be achieved by adding a term of regularisation in the algorithm. For one reason, the POD basis ensures that the time-average energy captured per mode (and hence, the coefficient of projection) decrease with mode number,
and we did not leveraged this property when minimising the cost function. Regularisation was not used in the present work, because our focus is on describing a new estimation algorithm and showing its performances "as is", without further fine tuning. Nonetheless, it clearly appears that, in the last series of tests above, the optimisation problem becomes ill constrained, since the error diverges (typical of over-fitting regimes) as we increase the number of parameters and/or use sparser observations.

## 5   Conclusions

In this study, a data-driven method is presented for the study of a wave scattered by a turbulent flow from numerical simulations. The approach we propose is to study the variability of the incoherent wave, resulting from quadratic interactions of these two motions, through the correlations between the slow wave modulation amplitudes and the jet components, both evolving at similar time scales. To perform this investigation, two decompositions for the wave were used: the EPOD method to extract wave structures correlated to the jet components, and the BBPOD technique capturing optimal modes of the broadband wave

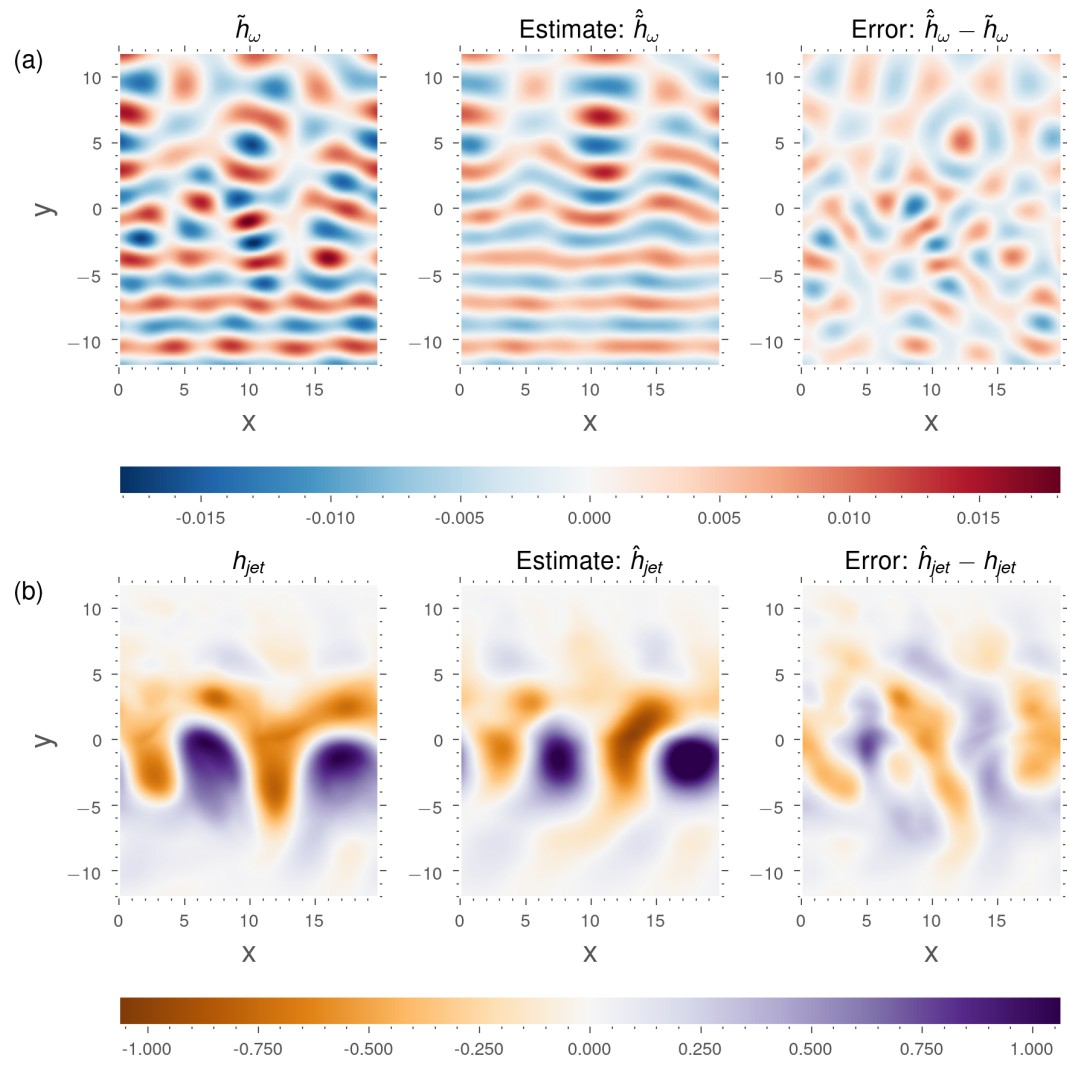

**Figure 12.** Estimate of the SSH contribution of the wave (**a**) and of the jet (**b**) for 12 modes from one sparse snapshot.





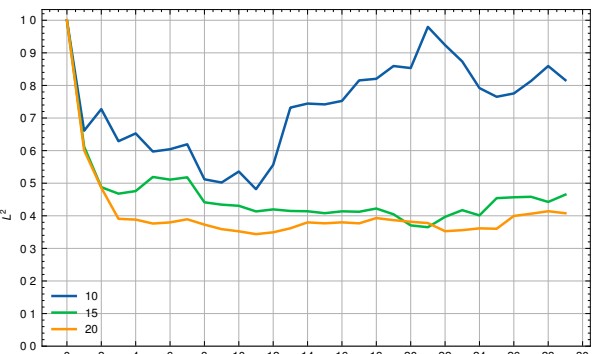 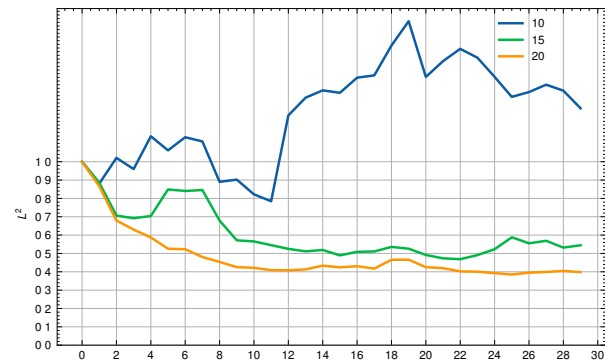

**Figure 13.** Averaged $L^2$ norm errors for the wave (**a**) and the jet (**b**) for sparse observations.

field. An explicit theoretical link between the EPOD modes of the wave and the POD modes of the jet is detailed when there is
negligible multiple scattering interactions. For an idealised rotating shallow water configuration where a wave is propagating
through a zonal jet, the numerical results confirm this connection. It has been demonstrated that for different waves and different
jets, the dominant modes of IGW are directly driven by those of the jet, resulting from single-scattering interactions with the
coherent wave. It has been shown that a large fraction of the energy of the incoherent wave is a standing wave resulting from

the deflection of the coherent wave by the meanders, according to this single interactions.

Based on this method, we derived a simple data assimilation scheme to demonstrate the capacity of such decompositions to
produce a simultaneous estimation of the jet and the wave from a single entangled observation of SSH. The EPOD method is
here meant to capture the IGW signal by correlation to the jet. In the case of a turbulent balanced flow dominating the SSH
signature, the EPOD method reveals its interest in order to estimate the weak IGW signal. Using only a single snapshot of

fully-observed SSH, the algorithm is able to estimate accurately all the components of the jet and the wave. The low-frequency
dynamics is optimally estimated (by property of the POD basis), and for the wave most of the wave patterns were captured,
accounting for approximately more than 70% of the total energy. However, the case of very incomplete observation shows that
high-order modes should be discarded and highlights the limitation of this simple assimilation scheme. As there is a coupling
with the jet components through the method, a proper estimate of the low-frequency dynamics is required to estimate the

IGW signal. To address this issue, appropriate regularisation terms could be added on this algorithm to better assimilate higher
modes. There is also a need to better quantify the part of the wave that is decorrelated from the jet, which is missing in the
algorithm.

Even though the method is for the moment limited to theoretical and idealised configuration, this provides some interesting
motivations for adapting it to more realistic cases. Indeed, the long revisit time of altimeters poses a problem to disentangle

the two dynamics from observations. Thus, the ability of the algorithm to produce accurate estimates from single snapshot
is a property that could be beneficial for such difficult configurations. Furthermore, as the method is meant to estimate IT in
region with strong mesoscale flows, it could also help addressing the issue of the disentanglement of the two dynamics in these



particular region, as in the Gulf Stream for instance. Egbert and Erofeeva (2021) also demonstrated that similar data-driven and modal decomposition technique could be envisaged to estimate IT from real observations. Yet, more work is required to test

the ability of the present modal decompositions to capture the two dynamics in realistic configurations. The method is firstly envisaged to be adapted on larger domains, by incorporating localisation technique, which is the object of an ongoing study.

*Code and data availability.* The subsampled time series of lowpassed filter and complex demodulated output of the RSW simulation W1 are provided at https://gitlab.inria.fr/imaingon/internal-tide-simulation.git, including codes to produce data and diagnostics.

## Appendix A: Equivalence between Broadband POD and SPOD

This appendix presents the algorithmic specifities of the common SPOD algorithm using the Welch method (Welch, 1967) to estimate the CSD, and its relation to the BBPOD algorithm, that is used in this study. For some well chosen parameters, we show that the two algorithms are equivalent.

The following proof is performed with discrete variables, considering a signal $x_t$ with time spacing $\Delta t$ and $t_k = k\Delta t$.

*Proof.* The complex demodulation writes:

$$
\begin{aligned}
\quad < x_t e^{-i\omega t} >_j &= \sum_{k=-m}^{m} b_k x_{j-k} e^{-i\omega t_{j-k}} \\
&= \sum_{k=j-m}^{j+m} b_{j-k} x_k e^{-i\omega t_k},
\end{aligned}
\tag{A1}
$$

where $(b_i)_{-m \leq i \leq m}$ are the discrete coefficients of the filter $< \cdot >$.

Besides, the principle of the Welch method is to subdivise $x_t$ into possibly overlapping blocks of size N with an overlap $N_o$. A Fast Fourier Transform is then performed on each windowed block to extract the Fourier component at the tidal frequency,

denoted $X_\omega^l$ where $l$ is the block index. So,

$$
X_\omega^l = \sum_{k=-N/2}^{N/2} x_{k+l(N-N_o)} W_k e^{-i\omega t_k},
\tag{A2}
$$

where $W_k$ is a window function defined on $[-N/2, N/2]$.

By changing variable $k' = k + l(N - N_o)$, it follows

$$
X_\omega^l = e^{i\omega t_{l(N-N_o)}} \sum_{k'=l(N-N_o)-N/2}^{l(N-N_o)+N/2} x_{k'} W_{k'-l(N-N_o)} e^{-i\omega t_{k'}}.
\tag{A3}
$$





Assuming that the window function is symmetric in the middle of each block (which is verified for most windows used in the literature), i.e $W_k = W_{-k}$, Eq.(A3) gives

$$X_\omega^l = e^{i\omega t_{l(N-N_o)}} \sum_{k'=l(N-N_o)-N/2}^{l(N-N_o)+N/2} W_{l(N-N_o)-k'} x_{k'} e^{-i\omega t_{k'}}. \tag{A4}$$

Finally, by choosing the window function as the filter coefficients, *i.e.* $W_k = b_k$ and $m = \frac{N}{2}$, relation Eq.(A1) yields:

$$X_\omega^l = e^{i\omega t_{l(N-N_o)}} < x_t e^{-i\omega t} >_{l(N-N_o)} . \tag{A5}$$

Consequently, up to a phase, the FFT of a block $l$ of size $N$ with overlap $N_o$ at $\omega$ corresponds to the complex demodulation of the signal at time $l(N - N_o)$. The phase shift cancels when computing the correlation over $N_b$ blocks :

$$\sum_{l=0}^{N_b} X_\omega^l X_\omega^{l^*} = \sum_{l=0}^{N_b} < x_t e^{-i\omega t} >_{l(N-N_o)} < x_t e^{-i\omega t} >_{l(N-N_o)}^* . \tag{A6}$$

Therefore the Welch method computed with parameters $(N, N_o, N_b, W)$ is equivalent as to compute the complex demodulation of the time series over $N_b$ snapshots sampled every $N - N_0$ and a filter chosen as the common window function $W$ (Hann, Hanning ...). □

This simple proof highlights the fact that the window function in the Welch method is playing the role of the filter in the BBPOD method. An advantage of the BBPOD algorithm for our study is that the frequency band covered by the window function is made explicit in order to extract the broadband wave field. In SPOD the paradigm is more to efficiently capture each harmonic separately, and to compute a basis at each frequency in the wave spectrum to capture all effects of incoherences. Using SPOD for spectral broadening problems would suggest to collect SPOD modes at neighbour frequencies an resort them in terms of energy content (as in Nekkanti and Schmidt (2021)). It would raise the issues of loss of orthogonality of the mode's family with respect to the spatial innerproduct, and to interpret properly the modes taking into account the potential spectral overlap induced by the Welch window, or at least to design it for purpose. We believe than that it makes it less adapted for studying broabdand fields as compared to the BBPOD method.

In the following is presented some complementary figures for the estimate part of Sect.4.3. First of all, a test has been performed in order to show the importance of informing the correlation between the jet contribution and the wave through EPOD in the estimation problem. A similar estimation than in Fig. 8 is performed, but independent coefficients $a_k$ and $b_k$ are sought for the estimation, instead being connected through EPOD. Results are shown in Fig. A1, and estimation errors are very large. Obviously, the optimal coefficients should lead to a lower error since some constraints are relaxed, but finding them becomes a hard task without physical information (provided by EPOD in our method), or advanced regularisation in the optimisation problem.

Secondly, estimates errors given a full observation are computed for the five simulations that are presented in Sect. 4, following algorithm 1. The maximal variance captured by the estimated wave field considering the first 30 modes is shown for each run in Table A1, as well as the fraction of the incoherent component that is recovered.



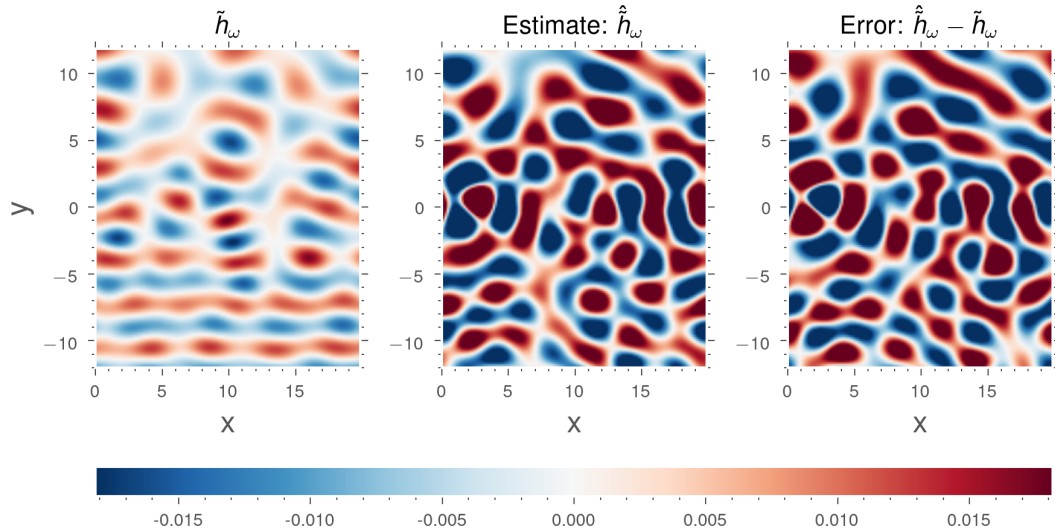

**Figure A1.** Estimation of the SSH contribution of the wave for the W1 run, computed from one snapshot, using 30 modes of the optimal basis of the jet and the wave vectors (POD for the jet, BBPOD for the wave). The BBPOD coefficients $b_n$ to recover the wave are computed according to the minimisation problem: $min_{(a_n,b_n)}\|\sum_{n=0}^{30} a_n\psi_n(\boldsymbol{q}_{jet})_h + \Re[b_n(\psi_{n,\omega})_h e^{i\omega t}] - h\|^2$.

| Estimates for each numerical simulation | | | | | |
|---|---|---|---|---|---|
| Mean-energy reduction | W1 | W2 | W3 | W4 | W5 |
| Total field | 73% | 64% | 82% | 88% | 75% |
| Incoherent part | 63% | 21% | 44% | 63% | 50% |

**Table A1.** Maximal values of variance recovered by the wave estimate for the first 30 modes, for each simulation given complete SSH observations. The values for the W1 run can be found in the corpus on Fig. 10 for 30 modes.

*Author contributions.* IM wrote the article, performed the numerical tests, and were implicated in the methodology of this study. GT and NL both supervised the global validation of the results and reviewed the article. They provided expertise on each section of the paper, and contributed to the formal analysis and to the global methodology of this article.

*Competing interests.* The authors declare that they have no conflict of interest.

*Acknowledgements.* N. Lahaye had support from the French research funding agency under the ModITO project (ANR-22-CE01-0006-01)
and from the TOSCA-ROSES SWOT project DIEGO. G. Tissot and N. Lahaye acknowledge support from the French National program LEFE (Les Enveloppes Fluides et l'Environnement).



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
