# Peer review of "Coupled estimation of incoherent internal tide and turbulent motions via statistical modal decomposition"

_EGUsphere, 2024_

## Referee Comment (RC2)

**Review of *Coupled estimation of incoherent inertia gravity wave field and turbulent balanced motions via modal decomposition**

The manuscript proposes a data driven method which can be used to extract the structures of a inertia-gravity wave field scattered by balanced eddies in the ocean. The authors extend the idea of proper orthogonal decomposition (POD) to broadband POD (BBPOD) and extended POD (EPOD). Using these two decompositions, the authors split the net flow into a geostrophically balanced coherent jet, fluctuations of the balanced jet and a wave field with a particular frequency $\omega$. They find that the two decompositions give the same qualitative results, and for low wave amplitudes and Rossby numbers, the decompositions do well. They also try to estimate the sea surface height (SSH) contributions of each mode for different wave forcings.

The study is essentially trying to make progress on identifying and separating fast inertia-gravity wave signals from sea surface height. This is an important and useful area of research, given the satellite data sets oceanographers work with and the state of the ongoing SWOT mission. However, I found the present manuscript to be lacking in multiple areas, in addition to mistakes in the writing and many relevant references missing. I recommend a major revision of the manuscript taking on board the comments detailed below.

**Major Comments**

1. Section 2.3 makes multiple statements without citing relevant references that can guide the readers for detailed calculations. The authors should carefully read through Dewar & Killworth 1995, Reznik et al. 2001, Thomas 2016, and Thomas 2023 and connect to these studies in section 2.3.

   As the authors say below line 115, one does not need to linearise the governing equations to get wave and balance equations. These calculations are discussed in detail in Dewar, W. K. & Killworth 1995, Reznik et al. 2001 and Thomas 2016 for O(1) amplitude waves. These calculations are generic in geophysical flow models, not specific to shallow water: see section 2 in Thomas 2023. Section 2.3 of the present manuscript can be much better written explaining these wave-balance splitting equations, thus informing a new reader important details and citing relevant references.

2. In this study, the authors conducted five numerical simulations of the rotating shallow water equations by focusing on the interaction between a plane wave and a zonal jet. The key parameters examined here include the temporal frequency

of the incoming wave, its direction of propagation, and the Rossby number of the turbulent jet. How do the energies of the incoming wave and the jet compare against each other? Can we treat the energy ratio between the wave and the jet as a parameter? Additionally, how would the results change if we alter the energy ratio of the incoming wave and the turbulent jet?

3. Equation 7 onwards: the authors have to explain whether there is any overlap between the components of the field after the decomposition. Additionally, time-filtering does not guarantee that energy of the different components separate. More discussion on the decomposition implemented numerically is needed for readers to follow.

4. Lines 140 - 145: Citations needed to justify the statement "We now make the assumption that the wave amplitudes evolve on the slow timescale compared to the wave time-period $2\pi/\omega$ (which can be formalized more rigorously by a WKB approach)". Sutherland 2014 and Fabrikant et al 1998 are relevant books that could be referenced here.

5. Line 151: Authors separate the wave solution into a coherent and incoherent part. However, the definitions of coherent and incoherent wave fields are not clearly described in the manuscript. Is the mean flow categorized as the coherent part and the fluctuating component regarded as the incoherent wave field? Please include an elaborate discussion.

6. Line 299: It is not clear why an eastward wind forcing is incorporated in the simulations. Is it possible that this wind forcing generates inertial oscillations? If so, why does the sea surface height (SSH) spectrum presented in Fig. 1 does not display a peak at the inertial frequency $f_0$?

7. Throughout section 3, the authors describe the details of the BBPOD and EPOD methods.However, the algorithm detailed in subsection 3.3 is extremely difficult to follow. The motivation or the outcome of each step in the algorithm is unclear. Specifically, it is not clear why a training stage is required as all equations are definitive and require no algorithmic learning. There is also no clarity on what is being trained. Additionally, I don't understand how a relative error is calculated as there is no mention of a ground truth in the algorithm.

8. Lines 295 - 300: The authors use periodic boundary conditions in both $x$ and $y$ direction. However, in the $\beta$ plane approximation, periodicity in the $y$ direction cannot be maintained. This is a contradiction. Are the authors setting $\beta = 0$? This is not explained in the manuscript.

9. Line 301: The authors used a radiative damping term to dissipate energy input from the wind and to obtain stationarity. Can the authors relate this damping term to any physical process or is it just a numerical technique which is implemented to facilitate numerical stability?

10. Line 304: What is a nudging layer? Line 325: What are the sponge regions? Why are these regions removed while calculating the POD and BPOD modes? These details have to be incorporated in the revision.

11. Line 370: Authors point out that in W5 run even though the patterns are similar to W1 and W2 runs, the meanders formed in W5 run have twice larger zonal wavelength and weaker energetic contribution. However, from the colorbar provided it is not clear how the energetic contributions are weaker in W5 run than W1 and W2 runs. I would like to see a difference plot between the runs that show that the energetic contribution are weaker in W5 than W1 and W2 runs?

12. In Fig. 5, the authors show the modal energy distribution of the incoherent wave field. It is not clear how exactly the incoherent energy is calculated. Line 353: Why is it expected that for the stronger jet the energy will be higher?

13. In Fig. 10 (caption is missing), the error almost saturate or decay very slowly in the range 20-30 modes. However, the authors claim that using a large number of modes render better estimates. Can the authors comment on the computing cost required for using large number of modes in performing the estimates? Do we expect better estimates if we use more than 30 modes? As far as I understand from Fig. 10, increasing the number of modes further will not provide better estimates as the error seems to have already saturated by N=30 modes.

14. The manuscript focuses exclusively on gravity waves, although this data driven method applies broadly to other similar wave scattering problems in the ocean. Scattering of small amplitude waves by a vortical flow or topography makes the wave field incoherent, leading to the excitation of new wave-modes with similar frequency as the parent wave. This has been seen for near-inertial waves (Danioux, & Vanneste 2016), acoustic waves (Thomas 2017) and surface waves (Thomas & Yamada 2018). The authors should broaden their summary section explaining the broader potential of the method they demonstrate for tides. Scattering of different small amplitude oceanic waves have been studied by Danioux, & Vanneste 2016, Thomas 2017 and Thomas & Yamada 2018 using asymptotic models. Since the present data driven method could be applied for these waves as well, the authors should discuss the broader applicability of their technique in the concluding section.

15. Following up on above comment, the present study addresses two dimensional scattering while oceanic waves can be scattered in three dimensions: see recent work by Kafiabad et al. 2019. How would the method the authors use in the present manuscript extend to three dimensions?

16. In the conclusion section the authors should also discuss in detail how the method would be challenged if O(1) waves were scattered by balanced flows, while being still in the small Rossby number limit. Some discussions on submesoscale interactions will also be useful.

**Minor comments**

1. Line 46: The abbreviation for spectral proper orthogonal decomposition is not specified here, but the authors use SPOD in Line 53.

2. Line 94: "A wave forcing term ...

3. Lines 95-100: The domain of the physical space is given as $\Omega \subset \mathcal{R}^2$. Later, however, the physical domain of the simulation is periodic (lines 295-300). This creates a confusion. Writing the equation in a conventional form (eg. the ones given in Vallis 2006) might be better.

4. In Eq 2, it is redundant to write $\Re$, instead using $cos(\omega t)$ makes the it clearer.

5. Lines 110-115: The details of the expectation operator are absent. It is unclear whether it is an expectation over multiple realisations, time, or physical space.

6. Line 105: A time scale separation is assumed between wave and balanced flow. This means that the authors are operating at mesoscales and not submesoscales, the latter scales having no wave-balance time scale separation. This needs to be acknowledged in the writing.

7. Lines 115-120: Adding "linearisation of Eq (1) about a balanced jet," instead of only "linearisation about Eq (1)," clears, at the onset, the steady state about which the linearisation is being performed.

8. Lines 145 - 150: The operator $R$ seems to be the Greens function of the linearized equation. Mentioning that is useful.

9. Line 156: " Finally, substracting ...", did the authors mean to write "substituting" in place of "substracting" ? Or is it "subtracting" ? The sentence is not clear and I would suggest rewriting this line.

10. Line 276: "...it could be subtracted... "

11. Line 322: It seems like this line is misplaced. Also no description on Fig. 2 is provided.

12. Line 341: Did the authors mean "..., which is pronounced for W3..."? In table 1, it is mentioned that $\omega = 3f_0$ for the W2 run. Also from Fig. 3 it is clear that it is in the W2 run, the north part of the domain exhibited a drop in the amplitude.

13. The labels on all the figures are very small, especially Figs. 4, 5, and 7. In Fig. 5 what is the x axis? Is it the number of modes N? Also, y axis label is not written correctly. Do the legends correspond to the different runs W1, W2, and so on? If yes, I suggest adding this as legends instead of showing the parameter values.

14. Figure 10 doesn't have a caption and the legends are missing. Thus the description of the figure given in lines 425-463 is very hard to follow. A revised figure with proper labels and legends should be provided.

15. $q$ in Eq 2 is used to denote the flow degrees of freedom whereas $q_{frc,w}$ is used to denote the wave forcing. This calls for confusion in the reader's mind. The flow is later decomposed into several components adding further subscripts to the flow degrees of freedom $q$, thereby increasing the confusion in reader's mind.

**References**

1. Sutherland, B. R. 2014, Internal Gravity Waves, Cambridge University Press.

2. Thomas, J. (2016): Resonant fast-slow interactions and breakdown of quasi-geostrophy in rotating shallow water, J. Fluid. Mech., 788: 492-520.

3. Dewar, W. K. & Killworth, P. D. 1995 Do fast gravity waves interact with geostrophic motions? Deep-Sea Res. I 42 (7), 1063–1081.

4. Thomas, J. (2023) Turbulent wave-balance exchanges in the ocean. Proc. R. Soc. A 479, 20220565.

5. Reznik, G. M., Zeitlin, V. & Ben Jelloul, M. 2001 Nonlinear theory of geostrophic adjustment. Part 1. Rotating shallow-water model. J. Fluid Mech. 445, 93–120.

6. Thomas, J. and Yamada, R. An amplitude equation for surface gravity wave-topography inter- actions, Phys. Rev. Fluids 3, 124802 (2018).

7. Danioux, E. & Vanneste, J. 2016 Near-inertial-wave scattering by random flows. Phys. Rev. Fluids 1, 033701.

8. Fabrikant, A. L., Stepanyants, Y. A. & Stepanyants, Y. A. 1998 Propagation of Waves in Shear Flows, World Scientific Series on Nonlinear Science Series A, vol. 18. World Scientific.

9. Thomas, J. New model for acoustic waves propagating through a vortical flow, J. Fluid Mech. 823, 658 (2017).

10. Kafiabad, H. A., Savva, M. A. & Vanneste, J. 2019 Diffusion of inertia-gravity waves by geostrophic turbulence. J. Fluid Mech. 869, R7.

---

## Referee Comment (RC3)

The manuscript entitled 'Coupled estimation of incoherent inertia gravity wave field and turbulent balanced motions via modal decomposition' investigates the identification problem of inertia gravity waves interacting with turbulent jets using POD and extended POD methods. The authors test their method on a rotating shallow water model. Given the two-point statistics of the turbulent field and the incoming wave, and assuming time scale and magnitude separations and strong correlation between the jet and the wave, they introduce a methodology that can distinguish the wave and the jet components from a single observation of the sea surface height. While I believe these assumptions significantly limit the method's applicability to more realistic scenarios, I find the methodology itself quite innovative. I appreciate that the authors managed to construct a model problem satisfying these assumptions.

My primary concern with the manuscript is its readability; the dense mathematical details required multiple readings to fully grasp. The text is mathematically quite involved, and I think, it can be simplified. For instance, the authors initially permit the complex amplitude of the inertia wave to vary over time, but later disregard this time dependency. I wonder whether it would be possible to reach the final resolvent equation by imposing classical Reynolds decomposition and linearization about the mean. Furthermore, the inconsistent use of terminology—particularly terms that carry different meanings across disciplines—complicates the understanding of the methodology. For instance, using an expectation operator, they decompose the jet into mean and fluctuating parts. But, the same expectation operator separates the wave into its coherent and incoherent parts.

I have listed specific areas that require further clarification; however, beyond addressing these, I recommend that the authors re-evaluate the entire text to enhance overall clarity. I believe the study merits publication, though its impact could be significantly enhanced with additional efforts to improve readability.

**Details:**

**lines 107 and 143:** I don't see why the authors define $\tilde{q}_\omega$ as a function of time, which is assumed to be constant later on anyway.

**lines 148-149:** The authors do not make any assumption about the frequency band they choose. How do they conclude about the fact that the resolvent operator is approximately constant? I would rephrase this sentence as '... can also be interpreted as assuming that the resolvent operator is approximately constant'

**line 151:** Why do the authors define this decomposition as coherent and incoherent while it is a Reynolds decomposition?

**Eq. (11):** If I understand it correctly, the first term in this equation is not fluctuating since the expectation operator (which amounts time or ensemble averaging, I guess) applies to the bilinear operator. So $\tilde{q}'_\omega$ is actually fluctuating about this term. If that is the case, I find it confusing that a prime term has a nonzero mean. Regarding the first use of the prime on line 140, I would relate it to quantities with zero mean.

**lines 167-170:** This spectral broadening effect applies to any term in eq. (11). But I think the authors particularly think of the term $B(q'_{jet}, \tilde{q}'_\omega)$. If that's the case, I think it would help if they explicitly stated that such as 'Taking $B(q'_{jet}, \tilde{q}'_\omega)$ for instance, ...'. It took me a while to figure out which incoherent component they were mentioning on line 170.

**Eq. (13):** How do the authors come up with this norm? Is it common in oceanography?

**Algorithm 1 – Training stage, last equation:** This assumes a strong correlation with h_omega and q_jet, as stated later in the text. Is not it a very limiting assumption? How realistic is it?

**Eq. (22):** How is this minimization achieved? The coefficients at the training stage are computed using the full vector q_jet, while in the minimization problem, only h is used. Does not this potentially cause a uniqueness issue? How do we know that the result of the minimization is unique?

**Section 4.1 -** A schematic showing the domain, discretization, the forcing and how the wave is introduced would be very helpful to visualize the test case.

**lines 359-360:** How are the EPOD modes calculated, applying POD to [q_jet, q_\omega] or first applying to POD to q_jet and then using the coefficients a_n(t) to reorganize q_\omega? How much error would be introduced by the former?

**line 366:** 'jet POD mode wave BBPOD mode' → 'jet POD mode and the wave BBPOD mode'

**lines 371-375:** Isn't it actually possible to calculate the single scattering term in eq. (19) and compare it against the EPOD modes?

**Figure 10.** The orange line is not mentioned in the caption.

**lines 443-444:** Why does the method fail for the W2 case?

**lines 450-451:** Is there a way to predict where the cut off the modes a priori?

---

## Author Comment (AC1)

**Review 1 : Coupled estimation of inertia-gravity wave and turbulent balanced motions**

Igor Maingonnat, Gilles Tissot, Noé Lahaye

December 21, 2024

We are grateful to the reviewer for carefully reading our manuscript and providing useful comments. We have substantially revised the manuscript following the Reviewer's suggestions, as well as 2 other Reviewers. In particular, the form of the manuscript (both regarding the language and organisation) has been reworked, which we believe have lead to substantial improvement of its clarity. Furthermore, we have make an effort in better discussing the potential realistic applications of the proposed methods. Below are our point-by-point responses to the Reviewer's comments (our answers appears in dark blue).

**Major points**

**Remark 1:** The form of the manuscript is insufficient at the moment and I fear it will adversely affect the impact this work may have. The manuscript is first of all lengthy and authors should strive to reduce its length. Here are some leads as to how to do that:

there are numerous repeats in transitions about what is about to be performed which are superfluous. L327-329 for instance is one example. Another example is L331-L334 where we have to wait three sentences before entering into the "meat" of the results, this is too long. Some optimization in the number of choice of figures could also help: Figure 5a and 5b are fairly redundant and their overall relevance may be questioned for the sake of conciseness; Figure 9 is barely discussed which should be taken as a sign it does not bring much and may be discarded.

Second, the level of language is low and substantially complicates the prompt understanding of the paper. I suggest resorting to AI (deepL, chatGPT), a professional editing service, or any other solution that may improve the flow of the manuscript.

**Answer:** We have reworded almost every sentences to improve the readability of the manuscript and remove language errors. We have also shortened some portions of the text, and removed redundancies. Following Referee's remarks, a particular attention has been give to headers of sections where redundancies were present, such as the beginning of Sect.3 "Methods".

In order to further reduce the length of the manuscript, as suggested, figure 9 has been removed from the manuscript with the corresponding lines L411-415. As suggested, Figure 5b is given in the appendix. We have also moved Figure 7 to the appendix, since it is similar to Figure 6. Figures 4(a), 4(b), 5(a) have also been reduced in size and merged into a single figure.

**Remark 2:** A second major concern is about the insufficient discussion regarding the applicability of the proposed method to more realistic configurations and the steps required to go there. The last paragraph of the conclusion is too light for that purpose. Potential questions I'd have liked to see addressed:

Is the level of mesoscale activity the sole parameter relevant to identify potential geographical areas of applications?

**Answer:** The mesoscale activity is the main parameter, but not the only one. Indeed, the EPOD technique considers instantaneous correlation between the jet and the wave. Therefore, it is well suited for configurations where 1) the signature of mesoscale dynamics is dominant over the internal tides, and 2) the domain is of limited extent – to minimize the impact of interactions at distance, which are associated with lagged (instead of instantaneous) correlation due to the finite wave propagation speed. Furthermore, the statistical convergence of POD modes (as a function of the number of samples) can become critical over large domains, which is identified as a limitation of our technique which deserves further work. Finally, a wave source must be well identified (either inside the domain or at the boundaries), such that the scattered field results from interactions between this predictable part and the mesoscale flow. The western boundary currents (*e.g.* Gulf Stream, Kuroshio, North Brazilian current) are example of configurations that match, at least partially, these criteria. We have enriched and clarified the discussion in the manuscript, also giving perspectives for how to deal with configurations that deviate from these restrictions.

Can we anticipate increased/decreased performances in realistic configuration given identified limitations? What observations and datasources could be used to evaluate the applicability of the method in realistic/operational configurations ? Or is more work required on the idealized side? And if yes to answer what questions? Do we have a grasp as to whether the methods will have to be adapted ? "localisation technique" is mentioned albeit with little detail and no references.

**Answer:** This method is based on statistical learning of the variability of waves and currents from a simulation. Consequently, variability that is not represented in realistic models can't be estimated from observations. In addition, a large number of samples is required to achieve statistical convergence of the POD/EPOD modes. Methods are needed to improve the convergence of statistical estimates – and localization techniques is a good candidate (the corresponding discussion has been enriched, with references added, in the conclusion). Another methodological constraint is due to the multiple constituents for the internal tide. As the different frequencies are very close with each others, it is necessary to be able to model the variability resulting from their coupling and from interactions with the mesoscale flow. These limitations are now identified as future work in the conclusion.

There are probably several possible paths, but we think that the most straightforward development is to test our method using outputs from realistic tide-resolving high-resolution numerical simulations (such as the mitGCM-base ECCO LLC4320 run, or HYCOM run). Preliminary tests using the NEMO-based eNATL60 simulation over the North-Atlantic have already shown some technical challenges (see our answer above) that need to be tackled. In addition, more work based on idealised experiments is also required to address the issues described above.

A more detailed comparison with similar approaches (Egbert and Erofeeva 2021 for instance) is also missing.

**Answer:** A more thorough comparison with Egbert & Erofeeva (2021) has been added in the new section 4.4.

**Minor points**

**Remark 1:** I suggest adjusting the title according: "Coupled estimation of incoherent internal tide and balanced turbulent motions via statistical modal decomposition". Modal decomposition is ambiguous with vertical modes used for internal tide descriptions, the method is targeting internal tide and not internal gravity waves in general.

**Answer:** The title has been changed as suggested by the reviewer. We also have removed the word "balanced" since it is another level of detail.

**Remark 2:** Terminology: the use of "internal tide" instead of "internal wave" seems more appropriate and I would recommend sticking to this choice throughout the manuscript.

**Answer:** This term has been modified in the entire manuscript.

**Remark 3:** Abstract: mention of the fact that this work is carried in an idealized configuration should come very early; sentences are too long; the first sentence is not necessary.

**Answer:** The abstract has been extensively modified for greater clarity. The first sentence has been removed as suggested by the reviewer, and the mention that the study is being carried out in an idealised configuration now appears earlier.

**Remark 4:** L28: "realistic" → idealized

**Answer:** The term realistic has been changed into idealized as suggested by the reviewer.

**Remark 5:** L29: "contribute significantly to" → enhances

**Answer:** "contribute significantly" has been changed to "enhances" as suggested by the reviewer.

**Remark 6:** L73: "the model used to investigate the dynamics" → the dynamical model used to investigate interactions between...

**Answer:** We changed the corresponding sentence, which has been shorten for conciseness and fluidity. The fact that the model is used to investigate the interactions is mentioned later (L.80 in the introduction of Sect.2.)

**Remark 7:** L111: "the potentially broaden spectrum" → typo?

**Answer:** This has been changed in broadband spectrum.

**Remark 8:** L112: "the fluctuations" → anomalies

**Answer:** For the sake of clarity, we have introduced the incoherent amplitudes as the residual of the coherent part. "Fluctuations" is used only for the low-frequency turbulent flow.

**Remark 9:** L117: "can be performed" → could have been performed

**Answer:** This part of the text has been removed during the shortening of the paper.

**Remark 10:** L145: can we expect that an inverse of the operator is always available ? If yes, this should be mentioned

**Answer:** We specified that the resolvent operator is well defined only if $-i\omega$ is not an eigenvalue of $L + B(\mathbb{E}[q_{jet}, \cdot])$. This is indeed not always the case in general. However, it is shown numerically that this operator is invertible in the thesis Maingonnat 2024. In addition, this operator has a countable number of eigenvalues, and due to the eddy diffusion term, the eigenvalues have a slight negative real part, while $-i\omega$ is pure imaginary. We do not consider $\omega = 0$ (eigenvalue associated with solutions close to geostrophic balance), where a standard POD is performed for the jet. This renders very unlikely to fall on an eigenvalue.

**Remark 11:** L145-150: physical implications for this are missing. If the domain was large enough, the impact of jet fluctuations should be delayed at a distance. Can this approach represent such situation? Such element are important to gauge the applicability of the method in other configuration (e.g. more realistic ones)

**Answer:** As suggested by the reviewer, we have added the physical interpretation of this hypothesis in section 2.3.2. The Reviewer is entirely correct in his remark on the impact of the size of the domain. Indeed, this hypothesis corresponds to a local scattering hypothesis and is relevant on small domains, which is now explicitly stated in the text. This is a limitation of our derivation: one main consequence is that interactions at distance are filtered out in the EPOD modes (since only instantaneous correlation are retained), and consequently, the estimation will not be able to represent the corresponding fraction (see also our response to the Reviewer's main remark 2). This is

identified as a perspective for further development of our method.

Additionally, we have identified that this physical assumption is associated with neglecting the time derivative of the slowly-varying amplitude term between equation (12) and (13) of the revised manuscript. Detailed comments have been provided L.176, end of Sect. 2.

**Remark 12:** L155: I find the presence of the nonlinear correction intriguing, maybe puzzling. Can we expect this correction to be substantial, e.g. shouldn't the correlation drop?

**Answer:** We interpret this term as a multiple scattering term, since it is the non-linear interaction between the incoherent wave component and the flow. This term may be small in the context of weak interactions and/or in localised regions where the scattered wave does not have the time to interact again with the flow. It may not be neglected in the general case.

**Remark 13:** L173: "showing that... as expected" : this statement does not follow scientific writing standards.

**Answer:** We removed this sentence. This whole discussion has been moved in the introduction.

**Remark 14:** L184: the SPOD acronym should be introduced as it is not described in the appendix

**Answer:** We introduced this acronym in the introduction.

**Remark 15:** L190: "The algorithm..." serious language issues here

**Answer:** This sentence has been fixed, thank you for pointing at it.

**Remark 16:** Eq 17 and elsewhere: it may be useful to retain spatial coordinates dependance at least sometimes. One may loose track about what depends on space and not at times.

**Answer:** The dependence of the field on $(x, y, t)$ has been added in Sect.3.1 and Sect.3.2.

**Remark 17:** L272 : "alternative method to using the geostrophic balance for BM..." this statement probes the question as to how far are velocity estimates from geostrophy and polarization relationships. You are not answering this in the manuscript which you may want to specify in order for the reader not remain in expectation.

**Answer:** Since this issue is not addressed in the manuscript, we have withdrawn this remark.

**Remark 18:** L282: " To our knowledge,..." I would strongly disagree, you need to be more specific.

**Answer:** We have modified this sentence: in the new version, we say that this is a difficult configuration for wave estimation and we cite Zaron (2017) L.282.

**Remark 19:** L331-332: need to report on values of alpha, hyperviscosity and all other parameters employed in the numerical simulations.

**Answer:** We have added the values of all the parameters, including hyperviscosity and $\alpha$, L.301.

**Remark 20:** L310: " a sufficient sampling " I do no understand what you mean – reformulate

**Answer:** We meant that the time series must be well resolved in time, so that the wave can be extracted from data by time-filtering (*i.e.* in order to prevent aliasing effect). This sentence has been reworked and merged with L.306 (of the preprint), which specifies the saving frequency of the ouputs.

**Remark 21:** L314: "less than 3 days" you need to specify at what latitude

**Answer:** We have the typical latitude at which this duration corresponds.

**Remark 22:** L314: "magnitude spectrum" don't you mean "power spectrum" instead?

**Answer:** Indeed, we have corrected "magnitude spectrum" to "power spectrum". We also have updated the figure by displaying the magnitude square, and considering a Welch method to estimate the power spectrum.

**Remark 23:** L314: the method employed for spectral estimation needs to be specified

**Answer:** The details of the Welch method have been provided L.316.

**Remark 24:** L320: "scale separation in amplitude" this is an awkward formulation

**Answer:** We have deleted this sentence and the paragraph, to avoid repetition, as the comparison with the Gulf Stream has already been made in Sect.3.

**Remark 25:** L324-325: This paragraph seems out of place

**Answer:** This was a typo and has been corrected in the manuscript.

**Remark 26:** L327-335: see major comment 1

**Answer:** The authors have tried to make these two paragraphs more precise and concise.

**Remark 27:** L335: "nudging" it would be more straightforward to talk about "wave forcing"

**Answer:** The term "nudging" has been replaced by wave forcing as suggested by the reviewer. In addition, we have replaced "nudging layer" by "sponge layer", which is a more proper formulation.

**Remark 28:** L339: "almost zero mean" this statement does not follow scientific writing standards.

**Answer:** We have fixed this formulation, and the whole sentence has been modified.

**Remark 29:** L346: "definition" → "construction"

**Answer:** We have reworded all the sentences and this passage has been deleted. We now mention that these modes are decorrelated from the coherent mode.

**Remark 30:** L346: "essentially non-zero" awkward formulation

**Answer:** This sentence has been removed to make the presentation of the modes more concise. In the revised version, we simply state that the modes consist of nearly-plane waves deflected by the jet in the upper part of the domain.

**Remark 31:** L349: "slight deviations to a single-mode structure" This needs to be more clearly reformulated.

**Answer:** This formulation has been clarified, thanks.

**Remark 32:** L351-L357: see major comment

**Answer:** This part of the manuscript has been reworked and shortened following the major comment.

**Remark 33:** L372-L373: could you use (19) to compute EPOD modes? If yes, has the correspondence been verified?

**Answer:** We have tested numerically the correspondence between EPOD and resolvent modes in this configuration. The results are in the PhD thesis of Igor Maingonnat entitled "Compréhension et modélisation de mécanismes non-linéaires dans l'océan : les interactions entre ondes internes et écoulement" (Chapter 4; PhD recently defended, the manuscript will be soon available via the French "thèse en ligne" repository). Moreover, it can be noted that a similar correspondence between resolvent analysis and EPOD has been established in Karban *et al.* (2022, 2023; cited in the paper) in the contect of turbulent channel and jet flows. Giving the corresponding details is out of the scope of the paper, but we have added the remark and the reference in the paper.

**Remark 34:** L381: "time evolution" of. . . ?

**Answer:** This sentence has been clarified.

**Remark 35:** L389: "stationary wave" Is "stationary" the most adequate term here, I fail to understand its precise meaning here.

**Answer:** Indeed, "stationary" is not the most adequate term. We replaced it by "standing wave", which describes the nodes and lobes formed by the superposition of two oppositely propagating plane waves. Thank you for this comment.

**Remark 36:** L390: "one third" → 30%

**Answer:** This has been modified.

**Remark 37:** L394-395: the second part of the sentence needs clarification

**Answer:** As this sentence was unclear and a matter of detail, and given the reviewer's major concerns about the length of the manuscript, we decided to delete this paragraph.

**Remark 38:** L405: "an accurate estimate" → "a visually accurate estimate"

**Answer:** This has been modified by "Qualitatively, the estimation is in good agreement with the reference".

**Remark 39:** L406: "a well identified structure" → vague statement

**Answer:** This description has been clarified.

**Remark 39:** L407-410: the description of the "naive" method needs to be improved. It may be also relevant to push this alternative approach towards the end of the section. Figure 9 is not particularly useful and may be skipped, color map is not adequate in any case.

**Answer:** Following the Reviewer's suggestion, Figure 9 has been removed. We have moved the description of the BBPOD-based method of the section. For the sake of conciseness, only key aspects are given in the main text, but a more detailed description is now given in Appendix.

**Remark 40:** L412: here or elsewhere you may want to specify that you could also have estimated wave energy fluxes

**Answer:** This is indeed a potential outcome of the method. We have added this remark in the conclusion and perspectives. Thank you for this suggestion.

**Remark 41:** L412: "more than 50% of the energy recovered at approximately each point. . . " this is not an adequate report of performance, you may want to report on an averaged or percentile value instead.

**Answer:** We have deleted this sentence as it was referring to Figure 9, which has been removed from the text at suggested by the Reviewer.

**Remark 42:** L416-417: specify that this is as a function of the number of modes

**Answer:** This detail has been added in the text.

**Remark 43:** L422 "The figure. . . " is this expected?

**Answer:** Indeed, since the jet is close to geostrophic balance, the SSH contains enough information to perform an accurate estimation of the velocity. We have added a remark in the text L.415.

**Remark 44:** L423: "For the wave..." I fail to see how you compute incoherent energy...make sure you explain this somewhere

**Answer:** We specified the calculation in section 4.3.1.

**Remark 45:** Figure 10: $x$ label and legend are missing on my computer.

**Answer:** This has been corrected.

**Remark 46:** Table A1: I would bring this table in the core of the manuscript and discuss it properly. The sensitivity to wave and jet properties is interesting. The table needs a bit of work (first lign is useless, make it clearer this is for wave part and/or add jet corresponding metrics)

**Answer:** The table has been improved in accordance to the suggestions.

**Remark 47:** L443: "Yet, this frequency..." please double check and specify latitude

We have double-checked and we specified the latitude at which this value corresponds.

**Remark 48:** L463: "of regularisation" you may want to specify "that penalizes higher mode amplitudes"

**Answer:** The formulation proposed by the Reviewer has been added in the text.

**Remark 49:** L479 "standing wave" is it the same as the earlier "stationary" wave? If yes you may want to align terminology and make sure it makes sense

**Answer:** Yes, this term has been used to designate the same thing. As the reviewer suggests, we have kept the term "standing wave" only.

**Remark 50:** L495-L501: see major comment 2

**Answer:** This discussion has been moved to the conclusion, and these points have been more detailed as suggested by the Reviewer in his major point 2.

**Remark 51:** L501: "localisation" reference missing

**Answer:** References have been added, thank you for pointing at this.

**Remark 52:** L504: "SPOD" acronym not specified I believe

**Answer:** We have added the information.

**Remark 53:** L540: this seems like a different subject from here on, so I would create another appendix section

**Answer:** We have created another appendix section.

---

## Author Comment (AC2)

**Review : Coupled estimation of inertia-gravity wave and turbulent balanced motions**

Igor Maingonnat, Gilles Tissot, Noé Lahaye

December 21, 2024

We would like to thank the Reviewer for agreeing to revise our manuscript. We have substantially revised the text, following the suggestions of the Reviewer and two other reviewers. In particular, the form of the manuscript (both in terms of language and organisation) has been reworked, leading to a significant improvement in its clarity. We have also made an effort to lighten the mathematical details. Below are our point-by-point responses to the reviewer's comments (our responses appear in dark blue).

**1   Major points**

**Remark 1**: Section 2.3 makes multiple statements without citing relevant references that can guide the readers for detailed calculations. The authors should carefully read through Dewar and Killworth 1995, Reznik et al. 2001, Thomas 2016, and Thomas 2023 and connect to these studies in section 2.3. As the authors say below line 115, one does not need to linearise the governing equations to get wave and balance equations. These calculations are discussed in detail in Dewar, W. K. and Killworth 1995, Reznik et al. 2001 and Thomas 2016 for O(1) amplitude waves. These calculations are generic in geophysical flow models, not specific to shallow water: see section 2 in Thomas 2023. Section 2.3 of the present manuscript can be much better written explaining these wave-balance splitting equations, thus informing a new reader important details and citing relevant references.

**Answer:** We have added the majority of the recommended references and linked them to our method. In particular, the complex amplitude of the wave can be seen as the dominant term of an asymptotic decomposition as presented in Reznik et al. 2001, Ward and Dewar 2010 for a RSW model. This has been stated in Sect.2.2. We should emphasize, however, that the main objective of the paper is to present data-driven methods for extracting and separating a mesoscale field and an incoherent wave field, and to estimate it from snapshots of observations. In this context, the wave/balance splitting equations provide some support to interpret these methods, but we do not aim at investigating the impact of the wave field on the mesoscale field (and we consider dynamical regimes in which the wave dynamics is essentially linear, see our answer to next question). Therefore, we do not provide an extensive list of references on these aspect. As the Reviewer mention, there is a vast corpus of papers on this subject which is much more relevant than our contribution. The suggested articles were mentioned at the end of Sect2.3.2.

**Remark 2**: In this study, the authors conducted five numerical simulations of the rotating shallow water equations by focusing on the interaction between a plane wave and a zonal jet. The key parameters examined here include the temporal frequency of the incoming wave, its direction of propagation, and the Rossby number of the turbulent jet. How do the energies of the incoming wave and the jet compare against each other? Can we treat the energy ratio between the wave and the jet as a parameter? Additionally, how would the results change if we alter the energy ratio of the incoming wave and the turbulent jet?

**Answer:** In this study, we deliberately placed ourselves in a low wave amplitude regime, which frees the study from the wave amplitude parameter by linearity, as remarked by the referee. Although this allows for a simpler dynamics, it is also associated with a more challenging retrieval of the wave field from SSH snapshots, as the mesoscale contribution dominates, and it is one of the main advantages of the propose decomposition method to be able to recover the wave field is such regime. Generalization to higher wave amplitude (whereby the wave/mesoscale amplitude becomes a parameter) is identified as a perspective for further work in the conclusion. Nevertheless, we propose below a discussion of the possible impact of higher wave amplitude on our results.

A first consequence of small wave amplitude is the absence of front sharpening, caused by wave-wave interactions, leading to the rise of harmonics. The second consequence is the absence of Reynolds stresses and then influence of the wave field on the jet. We agree with the referee that it may constitute an additional parameter, but we believe that the energy of the jet and the wave does not play a crucial role in the methodology, and in the fact that the primary interactions with the POD modes of the jet generate the most incoherent modes.

Concerning the rise of harmonics induced by front sharpening, they are filtered by the complex-demodulation step, which does not suppress energy transfers between scales, but still result in almost-sinusoidal waves.

Concerning the impact of wave amplitude on the system, Eq. (21), which shows that the EPOD modes are generated by interactions between the coherent part and the POD modes of the jet, can in all generality be deduced from an assumption of time scale separation. For example, for $O(1)$ waves, the only difference is the presence of Reynolds stresses in the evolution equation of the mean field (which is diagnosed from the non-linear simulation), as suggested in the references given by the reviewer. In the evolution equation for the wave, large wave amplitude may induce generalised Reynolds stresses which are related to energy transfer towards harmonics. This effect may affect the results, but we can remark that a part of these effects can be captured by the POD/EPOD procedure, since it is built using the outputs of the non-linear simulation. This constitutes a perspective for further investigations.

Preliminary tests in non-linear regime confirm that the results shown in the paper are not strongly affected. We have clarified these points in section 2.3.2. Discussion of the estimation results when the mesoscale flow is weak has been also added in the new section 4.4 "Discussion".

**Remark 3:** Equation 7 onwards: the authors have to explain whether there is any overlap between the components of the field after the decomposition. Additionally, time- filtering does not guarantee that energy of the different components separate. More discussion on the decomposition implemented numerically is needed for readers to follow.

**Answer:** In section 2, we have placed ourselves in an idealised situation where the frequency overlap between the jet and the wave is negligible, justifying the timescale separation assumption. This is verified in the spectrum of the solution Fig.1, showing a clear frequency separation between the jet and the wave. This allows a clean separation between the components by a simple time filtering. We agree with the reviewer that if these conditions are not met, such as in the presence of submesoscale activity, the separation between components is more challenging. We therefore mention throughout the manuscript that the jet is a mesoscale flow.

**Remark 4**: Lines 140 - 145: Citations needed to justify the statement "We now make the assumption that the wave amplitudes evolve on the slow timescale compared to the wave time-period $2\pi/\omega$ (which can be formalized more rigorously by a WKB approach)". Sutherland 2014 and Fabrikant et al 1998 are relevant books that could be referenced here.

**Answer:** We have added the proposed reference Sutherland (2014). However, we couldn't manage to have access to Fabrikant *et al.* (1998).

**Remark 5:** Line 151: Authors separate the wave solution into a coherent and incoherent part. However, the definitions of coherent and incoherent wave fields are not clearly described in the manuscript. Is the mean flow

categorized as the coherent part and the fluctuating component regarded as the incoherent wave field? Please include an elaborate discussion.

**Answer:** The definition of coherent and incoherent has been improved in section 2.2 "Complex amplitudes ansatz". In this context, the term *mean field* refers to the average of the low-frequency solution associated with the mesoscale flow, and the term *fluctuations* also refers to the fluctuations of the mesoscale flow. The coherent/incoherent parts of the wave fields corresopnd to the mean and fluctuations of the complex amplitude (which is extracted from the times series via complex demodulation).

**Remark 6:** Line 299: It is not clear why an eastward wind forcing is incorporated in the simulations. Is it possible that this wind forcing generates inertial oscillations? If so, why does the sea surface height (SSH) spectrum presented in Fig. 1 does not display a peak at the inertial frequency $f_0$ ?

**Answer:** Wind forcing is added to the simulation to maintain the Eastward jet throughout the simulation (L297-299). Otherwise, the jet energy would decrease due to the dissipation terms of the numerical simulation. The wind forcing is constant in time and and the Rossby number is moderate, therefore it does not generate inertial oscillations, as can be seen in the SSH spectrum in Figure 1.

**Remark 7:** Throughout section 3, the authors describe the details of the BBPOD and EPOD methods. However, the algorithm detailed in subsection 3.3 is extremely difficult to follow. The motivation or the outcome of each step in the algorithm is unclear. Specifically, it is not clear why a training stage is required as all equations are definitive and require no algorithmic learning. There is also no clarity on what is being trained. Additionally, I don't understand how a relative error is calculated as there is no mention of a ground truth in the algorithm.

**Answer:** The section "Algorithm" 3.3 has been clarified. In particular, it is now mentioned that the POD/EPOD modes are learned and that the POD coefficients are estimated by a least square algorithm. In the context of separating wave and currents from data, the knowledge of a model is a precious information, but cannot be employed directly. It can be employed for instance in a context of data assimilation by coupling model and data, through *e.g.* adjoint-based optimisation. In the present paper, the strategy is data driven and considers an a posteriori reduced-order modelling technique, which does not require a numerical implementation of the model, but at the price of a training procedure. We have clarified in the text the motivation and outcomes of the estimation procedure. The definition of the ground truth has been clarified as well in section 4.3 L387-388.

**Remark 8:** Lines 295 - 300: The authors use periodic boundary conditions in both $x$ and $y$ direction. However, in the $\beta$ plane approximation, periodicity in the $y$ direction cannot be maintained. This is a contradiction. Are the authors setting $\beta = 0$? This is not explained in the manuscript.

**Answer:** The treatment of inhomogeneity in the $y$ direction is performed with sponge layers, which is a standard technique in spectral methods. The linear dependence $\beta y$ in the Coriolis parameter is rendered periodic using a smooth recovery function acting only in the sponge layer. Furthermore, the wave field is generated (and re-absorbed at the North) in these sponge layers by nudging toward the incoming plane wave solution. These regions are not physical and are not included in the POD and BBPOD computation. These details have been added in the manuscript in section 4.1.

**Remark 9:** Line 301: The authors used a radiative damping term to dissipate energy input from the wind and to obtain stationarity. Can the authors relate this damping term to any physical process or is it just a numerical technique which is implemented to facilitate numerical stability?

**Answer:** The radiative damping term is mainly added in our simulation in order to ensure an energy balance and to obtain statistical stationarity of the solution. Indeed, some dissipation at large scale is required due to the inverse energy cascade in order to avoid energy accumulation. Such a term is often introduced in the literature to

model radiative damping, but in our study it is rather employed for numerical reasons than physical modelling. We have clarified this in the text.

**Remark 10:** Line 304: What is a nudging layer? Line 325: What are the sponge regions? Why are these regions removed while calculating the POD and BBPOD modes? These details have to be incorporated in the revision.

**Answer:** The term nudging layer has been changed to sponge layer, which is mentioned earlier in the manuscript to designate the same concept. As mentioned previously, the sponge regions are a numerical procedure to treat inhomogeneity with spectral methods, and clarifications have been provided in the revision.

**Remark 11:** Line 370: Authors point out that in W5 run even though the patterns are similar to W1 and W2 runs, the meanders formed in W5 run have twice larger zonal wavelength and weaker energetic contribution. However, from the colorbar provided it is not clear how the energetic contributions are weaker in W5 run than W1 and W2 runs. I would like to see a difference plot between the runs that show that the energetic contribution are weaker in W5 than W1 and W2 runs?

**Answer:** Indeed, the amplitude difference is only qualitatively visible by the difference of color saturation between figure 3 and C1 of the revised manuscript. (Formerly Figure 7 has been moved to the appendix (C1) for the sake of conciseness, following another Reviewer's suggestion.) We have added the values of eigenvalues, which corresponds to the corresponding energy captured by the mode, in the graphs figures 3, 5 and C1, in order to add the quantitative information.

**Remark 12:** In Fig. 5, the authors show the modal energy distribution of the incoherent wave field. It is not clear how exactly the incoherent energy is calculated. Line 353: Why is it expected that for the stronger jet the energy will be higher?

**Answer:** The definition of incoherent energy from BBPOD eigenvalues has been specified in the text in section 4.2.1 and in the legend. Figures 4 and 5 have been merged, and the normalised cumulative energy has been removed because it was redundant with the non-normalised version.
The authors have clarified the meaning of sentence L353, in L349-351 of the new version. Since the Rossby number is higher in the W5 simulation, the contribution of the non-linear terms between the wave and the jet, responsible for the generation of its incoherent part, is a priori more important as suggested by Eq.13.

**Remark 13:** In Fig. 10 (caption is missing), the error almost saturate or decay very slowly in the range $20 - 30$ modes. However, the authors claim that using a large number of modes render better estimates. Can the authors comment on the computing cost required for using large number of modes in performing the estimates? Do we expect better estimates if we use more than 30 modes? As far as I understand from Fig. 10, increasing the number of modes further will not provide better estimates as the error seems to have already saturated by N=30 modes.

**Answer:** We agree with the Reviewer that further increasing the number of modes leads to slight performance improvement, or even some increase of the error in situations of over-fitting (with partial observations). Even if the projection error is guaranteed to be reduced, this is not the case for the estimation step. It is often preferable to consider a more robust estimation with a reasonable number of modes kept than trying to estimate high-order modes leading at the end only to a slight improvement. The computational cost is very small in our case whatever the number of modes kept, since the minimisation problem is a simple least square of small dimension. The discussion of these aspects have been improved in the revised manuscript section 4.3.2.

**Remark 14:** The manuscript focuses exclusively on gravity waves, although this data driven method applies broadly to other similar wave scattering problems in the ocean. Scattering of small amplitude waves by a vortical flow or topography makes the wave field incoherent, leading to the excitation of new wave-modes with similar frequency as the parent wave. This has been seen for near-inertial waves (Danioux, and Vanneste 2016), acoustic

waves (Thomas 2017) and surface waves (Thomas and Yamada 2018). The authors should broaden their summary section explaining the broader potential of the method they demonstrate for tides. Scattering of different small amplitude oceanic waves have been studied by Danioux, and Vanneste 2016, Thomas 2017 and Thomas and Yamada 2018 using asymptotic models. Since the present data driven method could be applied for these waves as well, the authors should discuss the broader applicability of their technique in the concluding section

**Answer:** We thank the reviewer for this remark. Indeed, the method is not specific to IT, and can be extended to other wave-flow interaction configurations. However, our strategy is primarly to configurations where a coherent wave contribution is identified. This may be the case for instance for internal tides, barotropic tides, of in the context of aeroacoustics with a tonal incident wave interacting with a turbulent flow. Near inertial waves and surface waves should require either to develop other strategies or to identify clearly a coherent wave generated of incoming in the domain. We have enriched the conclusion is that sense.

**Remark 15:** Following up on above comment, the present study addresses two dimensional scattering while oceanic waves can be scattered in three dimensions: see recent work by Kafiabad et al. 2019. How would the method the authors use in the present manuscript extend to three dimensions?

**Answer:** This is indeed a promising perspective and extension of the present work. We mentioned in the conclusion that extended modes can be defined in 3D, but we recommend combining this with other decomposition techniques, such as vertical mode decomposition as in Kelly (2016). This allows to obtain a coupled shallow water system, where each vertical mode component is driven by a RSW-like equation, which are coupled by additional terms. This may constitute a direct application of EPOD-based estimations.

**Remark 16:** In the conclusion section the authors should also discuss in detail how the method would be challenged if O(1) waves were scattered by balanced flows, while being still in the small Rossby number limit. Some discussions on submesoscale interactions will also be useful.

**Answer:** We mention in the conclusion that a large amplitude incoherent wave remains correlated to the jet and therefore the extended modes remain a priori well defined. As stated in remark 2, the problem may remain very similar, but Reynolds stresses and generalised Reynolds stresses should be taken into account. A part of them are potentially accounted for implicitly through the BBPOD and correlation procedure, but this needs to be tested, and is a nice perspective of the present study.
The submesoscales does not respect the time-scale separation with the coherent wave. This would require further developments. It mainly challenges the ability to separate jet and wave components, where filtering is not possible anymore and models are likely to be necessary.

**2 Minor points**

**Remark 1:** Line 46: The abbreviation for spectral proper orthogonal decomposition is not specified here, but the authors use SPOD in Line 53.

**Answer:** The abbreviation SPOD has been added to the introduction.

**Remark 2:** Line 94: "A wave forcing term ...

**Answer:** This mistake has been corrected in the manuscript.

**Remark 3:** Lines 95-100: The domain of the physical space is given as $\Omega \in \mathbb{R}^2$ . Later, however, the physical domain of the simulation is periodic (lines 295-300). This creates a confusion. Writing the equation in a conventional form (eg. the ones given in Vallis 2006) might be better.

**Answer:** We have improved the definition by stating "... where $\Omega \in \mathbb{R}^2$ is the bounded spatial domain". This includes periodic domains.

**Remark 4:** In Eq 2, it is redundant to write $\Re$, instead using $\cos(\omega t)$ makes the it clearer.

**Answer:** As $\tilde{q}_\omega(t, x, y) \in \mathbb{C}^3$, which follows from the complex demodulation operation (Eq.5 in the revised version) Eq.2 is not equivalent to taking the $\cos(\omega t)$ instead of $\Re$.

**Remark 5:** Lines 110-115: The details of the expectation operator are absent. It is unclear whether it is an expectation over multiple realisations, time, or physical space.

**Answer:** The expectation operator is a time average. This has been specified in the new version in section 2.2 and in the methods in section 3.3.

**Remark 6:** Line 105: A time scale separation is assumed between wave and balanced flow. This means that the authors are operating at mesoscales and not submesoscales, the latter scales having no wave-balance time scale separation. This needs to be acknowledged in the writing.

**Answer:** As suggested by the reviewer, we have specified throughout the manuscript that the turbulent flow is a low-frequency mesoscale flow.

**Remark 7:** Lines 115-120: Adding "linearisation of Eq (1) about a balanced jet," instead of only "linearisation about Eq (1)," clears, at the onset, the steady state about which the linearisation is being performed.

**Answer:** The term linearisation has been removed and the decomposition of the solution $q_{jet} + \epsilon q_\omega$ has been given directly, to be more concise.

**Remark 10:** Lines 145 - 150: The operator $\boldsymbol{R}$ seems to be the Greens function of the linearized equation. Mentioning that is useful.

**Answer:** As suggested by the reviewer, the operator $\boldsymbol{R}$ is related to the Green's function. This has been mentioned in the revised version, and a references has been given.

**Remark 11:** Line 156: " Finally, substracting ...", did the authors mean to write "substituting" in place of "substracting" ? Or is it "subtracting" ? The sentence is not clear and I would suggest rewriting this line.

**Answer:** The authors did mean subtracting. This error has been corrected in the text.

**Remark 12:** Line 276: "...it could be subtracted... "

**Answer:** This paragraph has been removed from the manuscript in order to reduce its size, which was considered too long by the other reviewers.

**Remark 13:** Line 322: It seems like this line is misplaced. Also no description on Fig. 2 is provided.

**Answer:** This paragraph has been amended accordingly, with a fuller description of the figure as suggested by the reviewer.

**Remark 14:** Line 341: Did the authors mean "..., which is pronounced for W3.."? In table 1, it is mentioned that $\omega = 3f_0$ for the W2 run. Also from Fig. 3 it is clear that it is in the W2 run, the north part of the domain exhibited a drop in the amplitude.

**Answer:** Indeed, the authors wanted to refer to W2 and not W3. This has been corrected in the article.

**Remark 15:** The labels on all the figures are very small, especially Figs. 4, 5, and 7. In Fig. 5 what is the x axis? Is it the number of modes $N$? Also, $y$ axis label is not written correctly. Do the legends correspond to the different runs W1, W2, and so on? If yes, I suggest adding this as legends instead of showing the parameter values.

**Answer:** We have improved the figures.

**Remark 16:** Figure 10 doesn't have a caption and the legends are missing. Thus the description of the figure given in lines 425-463 is very hard to follow. A revised figure with proper labels and legends should be provided.

**Answer:** This has been corrected.

**Remark 17:** $q$ in Eq 2 is used to denote the flow degrees of freedom whereas $q_{frc,\omega}$ is used to denote the wave forcing. This calls for confusion in the reader's mind. The flow is later decomposed into several components adding further subscripts to the flow degrees of freedom $q$, thereby increasing the confusion in reader's mind.

**Answer:** The notation was indeed confusing, we have changed $h_{frc,\omega} \mapsto f_{h,\omega}$, $\vec{v}_{frc,\omega} \mapsto \vec{f}_{\vec{v},\omega}$.

---

## Author Comment (AC3)

**Review 3 : Coupled estimation of inertia-gravity wave and turbulent balanced motions**

Igor Maingonnat, Gilles Tissot, Noé Lahaye

December 21, 2024

We would like to thank the Reviewer for agreeing to revise our manuscript. We have substantially revised the text, following the suggestions of the Reviewer and two other reviewers. In particular, the form of the manuscript (both in terms of language and organisation) has been reworked, leading to a significant improvement in its clarity. We have also made an effort to lighten the mathematical details. Below are our point-by-point responses to the reviewer's comments (our responses appear in dark blue).

**1 Major points**

**Remark 1:** My primary concern with the manuscript is its readability; the dense mathematical details required multiple readings to fully grasp. The text is mathematically quite involved, and I think, it can be simplified. For instance, the authors initially permit the complex amplitude of the inertia wave to vary over time, but later disregard this time dependency. I wonder whether it would be possible to reach the final resolvent equation by imposing classical Reynolds decomposition and linearization about the mean. I have listed specific areas that require further clarification; however, beyond addressing these, I recommend that the authors re-evaluate the entire text to enhance overall clarity.

We have rewritten significant parts of the manuscript to improve its clarity and fluidity, and have clarified the mathematical developments in section 2, as suggested by the reviewer. In particular, the time dependence of the complex amplitude is discussed more explicitly in the revised version of the manuscript, and we have added the equation describing the evolution of the incoherent part in which its time derivative is not neglected (Eq. 12). The fact that the complex amplitude is time-dependent is a definition of the incoherent complex amplitude field (Eq. 4), which distinguishes it from the coherent amplitude – which is constant in time. This variation with time is due to scattering from the unsteady background flow and corresponds in spectral space to the spectral broadening mentioned in section 3. Although we neglect the time derivative in the subsequent developments, in order to build a predictive – and diagnostic – model of the incoherent wave, and to provide physical and mathematical interpretation of the results of the statistical decomposition methods presented in the manuscript, it must be kept in mind that the variability of the wave field across snapshots essentially arises from this time dependence. We have clarified to make this statement in the revised manuscript.

Concerning the simplification of the mathematical developments, classical resolvent analysis is indeed usually obtained by writing on the left-hand side the linearised system over the time-averaged mean flow and on the right-hand side a forcing term aiming at modelling the effect of the unresolved non-linearities. In our context, we identify the dominant nonlinearity, which is the interaction between the slow base-flow variations and the coherent wave. In this context, a careful writing of the interactions of the two components is necessary. This allows us to identify clearly the assumptions necessary to obtain the final resolvent equation (13). We have moreover clarified this last

step in the revised manuscript, highlighting that neglecting the time variations of the wave amplitude is related to considering a jet frozen with respect to the propagation time of the waves to travel the domain. This cannot be easily seen by performing approximations earlier in the derivation.

Furthermore, the inconsistent use of terminology—particularly terms that carry different meanings across disciplines—complicates the understanding of the methodology. For instance, using an expectation operator, they decompose the jet into mean and fluctuating parts. But, the same expectation operator separates the wave into its coherent and incoherent parts.

We have tried to standardise as much as possible the terminology of the article as suggested by the reviewer, and to make explicit the precise meaning of the terms used when there is a potential ambiguity across disciplines. An example is the change from internal waves and inertia gravity waves to internal tide, as this is the main subject of the study. The term coherent and incoherent part is standard in the context of internal tide dynamics, and indeed corresponds to a Reynolds decomposition of its complex amplitude, which have been clarified in section 2.2. The term fluctuation now refers exclusively to the low-frequency turbulent flow.

**2   Minor points**

**Remark 1:** I don't see why the authors define $\tilde{q}_\omega$ as a function of time, which is assumed to be constant later on anyway.

**Answer:** see our response to the main remark 1 above.

**Remark 2:** lines 148-149: The authors do not make any assumption about the frequency band they choose. How do they conclude about the fact that the resolvent operator is approximately constant? I would rephrase this sentence as '... can also be interpreted as assuming that the resolvent operator is approximately constant'.

**Answer:** The filter frequency cutoff corresponds to a typical width of the mesoscale frequency band, i.e. the most energetic low frequencies $< f_0$ (*c.f.* Fig.1). For simplicity, and because the spectral broadening of the wave results from the interaction with the broadband spectrum of the flow, the frequency band chosen is the same for the wave. A higher cut-off frequency has little effect on the results. In fact, a higher cut-off frequency will only introduce low-energy components into the jet or wave. A twice higher frequency cut-off was also tested, and POD/EPOD modes were the same.
Equation (13) of the revised manuscript, states that the incoherent wave component can be obtained applying the resolvent operator to the bilinear term $\boldsymbol{B}(\boldsymbol{q}'_{jet}, \mathbb{E}[\tilde{\boldsymbol{q}}_\omega])$ representing the interaction between the jet fluctuation and the coherent wave – here we focus on the single scattering term for simplicity. At first sight, it may appear to be obvious following a classical resolvent analysis point of view, since the resolvent operator maps the non-linear term and the response. However, it has to be remarked that the bilinar term is broad-band, and the resolvent operator is the one at the frequency $\omega$. Equation (13) can be interpreted by the fact that in the frequency band described above, the resolvent operator is almost constant, and can be used to predict the incoherent wave.

**Remark 3:** line 151: Why do the authors define this decomposition as coherent and incoherent while it is a Reynolds decomposition?

**Answer:** The terms "coherent" and "incoherent" are standard in the physical oceanography community. We do not use the term "mean" and "fluctuation" because it corresponds to a Reynolds decomposition of the complex demodulated wave amplitude (not directly the wave field, which obviously gives 0 upon averaging). We have tried to remove this ambiguity by clarifying their precise definition in section 2.2.

**Remark 4:** Eq. (11): If I understand it correctly, the first term in this equation is not fluctuating since the

expectation operator (which amounts time or ensemble averaging, I guess) applies to the bilinear operator. So $\tilde{q}'_\omega$ is actually fluctuating about this term. If that is the case, I find it confusing that a prime term has a nonzero mean. Regarding the first use of the prime on line 140, I would relate it to quantities with zero mean.

**Answer:** Eq. (11) is now Eq. (13). According to the definition Eq. (4), $\tilde{q}'_\omega$ has zero mean, justifying the use of the prime notation. In addition, taking the expectation operator of the right hand side of Eq. (13) leads to 0, and this equation is therefore consistent with the definition (Eq. 4). Under ergodicity hypothesis, Eq.12 is also consistent with the definition.

**Remark 5:** lines 167-170: This spectral broadening effect applies to any term in eq. (11). But I think the authors particularly think of the term $B(q'_{jet}, \tilde{q}'_\omega)$. If that's the case, I think it would help if they explicitly stated that such as 'Taking $B(q'_{jet}, \tilde{q}'_\omega)$ for instance, ...'. It took me a while to figure out which incoherent component they were mentioning on line 170.

**Answer:** This part of the text has been removed in order to shorten the paper. The idea of convolution is now introduced in the introduction section, at a less detailed level. We hope that the idea of spectral broadening is more clearly presented in the revised version.

**Remark 6:** Eq. (13): How do the authors come up with this norm? Is it common in oceanography?

**Answer:** This norm corresponds to the total energy, *i.e.* the sum of kinetic and potential energy, for a small perturbation. This has been clarified in the revised version of the manuscript.

**Remark 7:** Algorithm 1 – Training stage, last equation: This assumes a strong correlation with $h_\omega$ and $q_{jet}$, as stated later in the text. Is not it a very limiting assumption? How realistic is it?

**Answer:** We have detailed the meaning of this sentence in the new section 4.4 "Discussion" where we discuss the limit of the method. We have connected it with sentence L270-271 of the preprint "However, the part of the wave that is completely decorrelated from the jet is not estimated in this algorithm, as it is a quantity that is more difficult to quantify". We point out in section 4.4 that this is not such a limiting hypothesis since in a large part of the ocean regions the incoherent wave is generated by the (bilinear) interaction of the wave with the currents, and is therefore correlated with these currents.

**Remark 8:** Eq. (22): How is this minimization achieved? The coefficients at the training stage are computed using the full vector $q_{jet}$, while in the minimization problem, only $h$ is used. Does not this potentially cause a uniqueness issue? How do we know that the result of the minimization is unique?

**Answer:** The minimisation is solved by a least square problem, with POD coefficients of the jet as control parameters (clarified in the text). In all our tests, there are more observations than modes sought. The least square problem is then unique. However as there are no regularisation term, the system may be subject to overfitting, as it is discussed in section 4.3.1.

**Remark 9:** Section 4.1 - A schematic showing the domain, discretization, the forcing and how the wave is introduced would be very helpful to visualize the test case.

**Answer:** We have added the sponge layers to this figure, delimited by dotted lines. We have also added grey areas corresponding to "relaxation" zones where the variables are damped to 0 during simulations, to ensure the periodicity of the solution calculated by the spectral method. In these sponge zones, we have added the region where the wave is generated.

**Remark 10:** lines 359-360: How are the EPOD modes calculated, applying POD to $[q_{jet}, q_\omega]$ or first applying to POD to $q_{jet}$ and then using the coefficients $a_n(t)$ to reorganize $q_\omega$? How much error would be introduced by the

former?

**Answer:** In practice, the EPOD modes are calculated by performing a POD with a semi-norm (*i.e.* with a POD on $(q_{jet}, \tilde{q}_\omega)$, with no weight on $\tilde{q}_\omega$). This has been more precisely stated in Sect.3.2. This is equivalent as calculating POD on $q_{jet}$ and using the coefficients to calculate the EPOD modes, as presented in the text formally. This has also been verified numerically. We also checked that putting weights on the wave did not change the estimate, due to the small amplitude of the wave. The paragraph mentioning this comparison has been removed from the manuscript to lighten the text.

**Remark 11:** line 366: 'jet POD mode wave BBPOD mode' → 'jet POD mode and the wave BBPOD mode'

**Answer:** This sentence has been modified.

**Remark 12:** lines 371-375: Isn't it actually possible to calculate the single scattering term in eq. (19) and compare it against the EPOD modes?

**Answer:** This can be verified numerically. These results are available in the PhD thesis of Maingonnat (2024), chapter 4, with a homogeneous direction. The reference has been added in the text. We do not show the results here because the methodology is very different from that used in the article.

**Remark 13:** Figure 10. The orange line is not mentioned in the caption.

**Answer:** It is now mentioned in the revised manuscript.

**Remark 14:** lines 443-444: Why does the method fail for the W2 case?

**Answer:** The loss of coherence is more intense for W2 and the resulting incoherent wave field is more complex. This was detailed in Ward and Dewar (2010) when the frequency becomes high. For example the contribution of multiple scattering term is greater for small scale waves. This also can be seen in figure 5, where the drop of amplitude of the coherent part in the north of the domain is very clear. Furthemore there is a larger scale separation between the domain size and the wavelength. This may increase the need for localisation, and the difficulty of a reduced basis to represent a more complicated solution. These effects combined may cause the loss of estimation performances.
These explanations are conjectures, and deeper analyses are required to clarify this point. We have noticed in section 4.3.1 the fact that it is still an open question.

**Remark 15:** lines 450-451: Is there a way to predict where the cut off the modes a priori?

**Answer:** Usually, the number of modes is selected *a posteriori*. Some *a priori* standard criteria exist such as Baysian information criterion (BIC) or Akaike information criterion (AIC). They account for the number of samples and the number of degree of freedom of a model to predict if we are likely in an overfitting regime. These were not tested in our configuration, since the method is computationally fast and allow easily an *a posteriori* choice.

---

## Referee Report (RR1)

**Review of *Coupled estimation of incoherent internal tide and turbulent balanced motions via statistical modal decomposition**

The revised manuscript is much improved than the first submission, making connections with several related references and tighter writing. However, some of the important issues pointed out in the first review remain in the present submission as well. I recommend another revision taking into account the comments below.

1. Below equations 1: The $\beta$ term can be retained in the equations only if the domain is unbounded in the $y$-direction with solutions decaying far off in $y$. This is essentially achieved with sponge layers in the present work. Given this, it is incorrect to say the domain is periodic in both directions. Periodicity implies that it is some what a finite domain with solutions repeating over and over again, while solutions have to decay in $y$ over large distances.

2. Above equation 3: Even in this revision the writing is sophisticated with the unnecessary expectation operator. In line 116 the authors say that the expectation operator is equivalent to a time average. Much later, in line 237 the authors say that "the expectation operator is a time average in this study". there is no reason to introduce an ambiguous expectation operator if it is identically a time-averaging operator. The former can confuse a lot of readers and deter them away from the main message of the manuscript.

3. Line 161: how do you subtract an equation TO another equation? Line 466: now should appear instead of know.

   There are a lot of typos and English errors throughout the manuscript. I suggest getting an online editor to fix all the mistakes in writing.

4. Line 300: why is hyperdissipation in this fractional form?

5. Several figures (1, 7, 10) have no captions, which makes it difficult for a reader to processes the information in these figures when looking at them.

6. The paragraph bellow line 506 can be better written with references. As mentioned in my previous review, surface waves, acoustic waves, and near-inertial waves are scattered by eddies and topography in the ocean and the method developed in the manuscript could be useful for those problems as well. The authors don't seem to appreciate how far reaching their method is with regards to all these different waves, as seen from their response to the comment I wrote. The authors should read the references I mentioned; see for example figure 3 in Daniuox and Vanneste

2016. Similar scattering of acoustic and surface waves can be seen in the references I mentioned. Including those references and making connections will lead to a better summary paragraph regarding the broader outreach of the method used in this manuscript. Readers will appreciate getting an idea that the data-driven method developed in this manuscript can be used for other wave scattering problems mentioned in the referenced papers.

7. After rewriting the above mentioned paragraph with references, it is better to modify the last line of the paragraph into a new paragraph. Three-dimensional scattering requires a bigger discussion and the last part where the authors say vertical mode decomposition can be used is too short and unclear. Similar to the comment above, I suggest that the authors write a bit more in detail keeping in mind readers who might be unfamiliar with what they are trying to say.

8. I suggest shortening the manuscript title to make a more apt and catchy one.

---

## Author Response (AR2)

**Review : Coupled estimation of incoherent internal tide and turbulent motions via statistical modal decompositions**

Igor Maingonnat, Gilles Tissot, Noé Lahaye

February 6, 2025

I have one minor question that I kindly ask the authors to respond to either in the final version of the paper or in a short note sent to the editor.

**Question** They investigate the effect of partial measurements by taking strips of the data with increasing thickness. Why did the authors choose this form of partial measurement over, for instance, pointwise measurements distributed over the domain? Is there a practical reason behind this? And, how would the predictions change in the case of pointwise measurements with increasing density instead of strips with increasing thickness?

**Answer:** The authors indeed considered SSH bands to model partial observations instead of a cloud of points. These SSH bands are supposed to represent observations from altimetry data. The authors chose these type of data as it is widely discussed and used in the literature for estimating turbulence and tides, and the launch of the SWOT satellite reinforces interest in this type of observation. Nonetheless, it could indeed have been possible to consider an ensemble of points in order to model observations from tracers, collected by argo data for example. We believe that a possible advantage of a cloud of points is that it provides more homogeneous coverage of the domain, compared with SSH strips where some areas are completely masked. This may attenuate errors in the case of very low domain coverage (e.g. 10% coverage in Fig.10). For realistic applications, knowledge of both types of observation is essential.

1. Below equations 1: The $\beta$ term can be retained in the equations only if the domain is unbounded in the y-direction with solutions decaying far off in $y$. This is essentially achieved with sponge layers in the present work. Given this, it is incorrect to say the domain is periodic in both directions. Periodicity implies that it is some what a finite domain with solutions repeating over and over again, while solutions have to decay in $y$ over large distances.

   **Answer:** We have deleted the two sentences below equation 1 L99-100 indicating that the domain is periodic to avoid any confusion on this point. Instead, we explain more clearly in section 4.1 how we treat numerically the upper/lower boundary conditions. We agree that using a periodic domain with a beta plane Coriolis parameter is indeed contradictory, since it would be associated with a discontinuous sawtooth profile for the latter. Using a truly bounded domain in the $y$-direction is, however, absolutely fine. Here, we employ a widely used strategy (especially for wave problems, *e.g.* [1, 2]) that consists of using absorbing sponge layers at both boundaries combined with a periodic domain. Following this approach, two subdomains are defined: the inner subdomain has an unaltered physics (with no additional damping and linear Coriolis parameter) while the outer domain has a modified physics with a nudging toward a desired state. In the sponge layer, the Coriolis parameter does not follow the beta-plane approximation, but instead decreases to match the value at the other endpoints (thus leading to a smoothing of the discontinuity in the original profile, and resolving the initial contradiction between using a beta-plane Coriolis parameter and periodic domain). This allows us to use periodic boundary conditions for the numerical simulations, although the dynamics across the sponge layers (that is, over repetitions of the domain) are not connected with each other because the nudging terms damp any dynamical features that comes from the inner subdomain. Thus, the numerically resolved domain is indeed doubly periodic, thanks to the addition of sponge layers in the $y$-direction which delimit a physical domain which is bounded in that direction.

2. Above equation 3: Even in this revision the writing is sophisticated with the unnecessary expectation operator. In line 116 the authors say that the expectation operator is equivalent to a time average. Much later, in line 237 the authors say that "the expectation operator is a time average in this study". there is no reason to introduce an ambiguous expectation operator if it is identically a time-averaging operator. The former can confuse a lot of readers and deter them away from the main message of the manuscript.

   **Answer:** As suggested by the reviewer, we have replaced the expectation notation $\mathbb{E}$ with a time-averaged notation $\langle \cdot \rangle$. The filtering operation has been replaced by $\bar{\cdot}$.

3. Line 161: how do you subtract an equation TO another equation? Line 466: now should appear instead of know.

   There are a lot of typos and English errors throughout the manuscript. I suggest getting an online editor to

fix all the mistakes in writing.

**Answer:** We thank the reviewer for pointing out these errors. We have corrected them. We have tried to correct these language errors as far as possible.

4. Line 300: why is hyperdissipation in this fractional form?

**Answer:** Hyperviscosity is in a fractional form following case 7 in Table 1 of Ochoa et al (2011), which is referenced in the manuscript. This form ensures conservation of angular momentum and energy dissipation for a RSW model in the f-plane. It has been scaled according to our scaling.

5. Several figures (1, 7, 10) have no captions, which makes it difficult for a reader to processes the information in these figures when looking at them.

**Answer:** The captions are correct on our submitted version.

6. The paragraph bellow line 506 can be better written with references. As mentioned in my previous review, surface waves, acoustic waves, and near-inertial waves are scattered by eddies and topography in the ocean and the method developed in the manuscript could be useful for those problems as well. The authors don't seem to appreciate how far reaching their method is with regards to all these different waves, as seen from their response to the comment I wrote. The authors should read the references I mentioned; see for example figure 3 in Daniuox and Vanneste i 2016. Similar scattering of acoustic and surface waves can be seen in the references I mentioned. Including those references and making connections will lead to a better summary paragraph regarding the broader outreach of the method used in this manuscript. Readers will appreciate getting an idea that the data-driven method developed in this manuscript can be used for other wave scattering problems mentioned in the referenced papers.

**Answer:** The paragraph has been modified to include citations recommended by the reviewer as well as certain links with our study. The estimation procedure employed in our study relies on the knowledge of a coherent field, in order to identify the relevant non-linearities through complex demodulation. Such a feature is not always available in the mentioned configurations, rendering impossible the exact application of the estimation strategy of our paper. However, we do believe that the concept of EPOD on complex demodulated fields may find nice applications in these configurations, such as near inertial waves, provided that a separation between the flow and wave fields is possible. We modified the conclusion in this sense, and we thank the reviewer for his enthusiasm regarding the method. Besides, the case of scattering by a topography does not seem to fall within the scope of application, since it is necessary for the scatterer to vary over time in order to compute its statistics.

7. After rewriting the above mentioned paragraph with references, it is better to modify the last line of the paragraph into a new paragraph. Three-dimensional scattering requires a bigger discussion and the last part where the authors say vertical mode decomposition can be used is too short and unclear. Similar to the comment above, I suggest that the authors write a bit more in detail keeping in mind readers who might be unfamiliar with what they are trying to say.

**Answer:** We have created a new paragraph, where explain more in detail how our method can be extended to 3D scattering from a base of vertical modes. We have also provided examples of 3D models projected onto these bases for the study of internal tides.

8. I suggest shortening the manuscript title to make a more apt and catchy one.

**Answer:** As suggested by the reviewer we have shortened the title "Coupled estimation of incoherent internal tide and turbulent motions via statistical modal decompositions" in Coupled estimation of internal tides and

turbulent motions via statistical modal decompositions". Note that as it has already been rebuilt at the suggestion of another reviewer, we prefer not to modify it further.

**References**

[1] Brès GA, Jordan P, Jaunet V, et al. Importance of the nozzle-exit boundary-layer state in subsonic turbulent jets. Journal of Fluid Mechanics. 2018;851:83-124. doi:10.1017/jfm.2018.476

[2] Sasaki, K., Tissot, G., Cavalieri, A.V.G. et al. Closed-loop control of a free shear flow: a framework using the parabolized stability equations. Theor. Comput. Fluid Dyn. 32, 765–788 (2018). https://doi.org/10.1007/s00162-018-0477-x